

# Study of the main processes driving atmospheric CH₄ variability in a rural Spanish region

Claudia Grossi[a,1], Felix R. Vogel[b], Roger Curcoll[a,2], Alba Àgueda[a,3], Arturo Vargas[c], Xavier Rodó[a,d,4], Josep-Anton Morguí[a,e,2]

[a] *Institut Català de Ciències del Clima (IC3), Barcelona, Spain*

[b] *Laboratoire des Sciences du Climat et l'Environnement (LSCE-IPSL), CEA-CNRS-UVSQ, Université Paris-Saclay, Gif-sur-Yvette, France*

[c] *Institut de Tècniques Energètiques (INTE), Universitat Politècnica de Catalunya (UPC), Barcelona, Spain*

[d] *Institució Catalana de Recerca i Estudis Avançats (ICREA), Barcelona, Spain*

[e] *Departament Biologia Evolutiva, Ecologia i Ciències Ambientals, Universitat de Barcelona (UB), Barcelona, Spain*

*Present addresses:*

[1] *Institut de Tècniques Energètiques (INTE), Universitat Politècnica de Catalunya (UPC), Barcelona, Spain*

[2] *Institut de Ciència i Tecnologia Ambientals (ICTA), Universitat Autònoma de Barcelona (UAB), Cerdanyola del Vallès, Spain*

[3] *Centre d'Estudis del Risc Tecnològic, Universitat Politècnica de Catalunya (UPC) - BarcelonaTech, Barcelona, Spain*

[4] *CLIMA2, Climate and Health Program, ISGlobal (Barcelona Institute of Global Health), Barcelona, Spain*

*Correspondence to*: Claudia Grossi (claudia.grossi@upc.edu), Felix Vogel (felix.vogel@lsce.ipsl.fr)

**Abstract.** Atmospheric concentrations of the two main greenhouse gases (GHGs), carbon dioxide ($CO_2$) and methane ($CH_4$), are continuously measured since November 2012 at the Spanish rural station of Gredos (GIC3), within the climate network ClimaDat, together with atmospheric radon ($^{222}Rn$) tracer and meteorological parameters. The atmospheric variability of $CH_4$

concentrations measured from 2013 to 2015 at GIC3 has been analyzed in this study. It is interpreted in relation to the variability of measured $^{222}Rn$ concentrations, modelled $^{222}Rn$ fluxes and modelled heights of the planetary boundary layer (PBLH) in the same period. In addition, nocturnal fluxes of $CH_4$ were estimated using two methods: the Radon Tracer Method (RTM) and one based on the EDGARv4.2 bottom-up emission inventory. Both previous methods have been applied using the same footprints, calculated with the atmospheric transport model FLEXPARTv6.2.

Results show that daily and seasonal changes in atmospheric concentrations of $^{222}Rn$ (and the corresponding fluxes) can help to understand the atmospheric $CH_4$ variability. On daily basis, the variation in the PBLH mainly drives changes in $^{222}Rn$ and $CH_4$ concentrations while, on monthly basis, their atmospheric variability seems to depend on changes in their emissions. The median value of RTM based methane fluxes (FR_CH₄) is 0.17 mg $CH_4$ m⁻² h⁻¹ with an absolute deviation of 0.08 mg $CH_4$ m⁻² h⁻¹. Median methane fluxes based on bottom-up inventory (FE_CH₄) is of 0.32 mg $CH_4$ m⁻² h⁻¹ with an absolute deviation of 0.06 mg $CH_4$ m⁻² h⁻¹. Monthly FR_CH₄ flux shows a seasonality which is not observed in the monthly FE_CH₄ flux. During January-May FR_CH₄ fluxes present a median value of 0.08 mg $CH_4$ m⁻² h⁻¹ with an absolute deviation of 0.05 mg $CH_4$ m⁻² h⁻¹ and a median value of 0.19 mg $CH_4$ m⁻² h⁻¹ with an absolute deviation of 0.06 mg $CH_4$ m⁻² h⁻¹ during June-December. This seasonal doubling of the median methane fluxes calculated by RTM at the GIC3 area seems to be mainly related to

the alternate presence of transhumant livestock in the GIC3 area. The results obtained in this study highlight the benefit of applying independent RTM to improve the seasonality of the emission factors from bottom-up inventories.

Keywords: methane, flux, radon, atmosphere, livestock, EDGAR, FLEXPART.

## Introduction

The impact of the atmospheric increase of the greenhouse gases (GHGs) on climate change is well known (IPCC, 2013). GHGs emissions, due to natural as well as anthropogenic sources, are currently estimated and reported to the United Nations Framework Convention on Climate Change (UNFCCC). A good understanding of the underlying processes causing the emissions can help in the implementation of emission reduction strategies. Among the GHGs covered under the UNFCCC framework, methane ($CH_4$) is the second most important anthropogenic GHG. The atmospheric concentration of $CH_4$ has substantially changed since pre-industrial times from a global average of 715 ppb to more than 1774 ppb (IPCC, 2013). Today the contribution of $CH_4$ related to anthropogenic activities in the atmosphere represents about 25% of the total additional anthropogenic radiative forcing (IPCC, 2013). However, $CH_4$ has a

relatively short lifetime in the atmosphere (~ 9 years) and this makes it relevant to define immediate and efficient emission reduction strategies (Prinn et al., 2000). Particularly, in Spain man-made methane emissions are mainly due to enteric fermentation (31%), management of manure (20%), and landfills (36%) (WWF, 2014; MMA, 2016). The remaining methane contributions in Spain are due to rice cultivation (e.g. Àgueda et al., 2017), coal mining, leaks in natural gas infrastructures and waste water treatment. The $CH_4$ emission due to enteric fermentation related to livestock is directly linked to the number of animals of each type/breed of cattle, their age, their diet and environmental conditions (MMA, 2016). Spanish $CH_4$ emissions for 2014 due to enteric fermentation were estimated to be of 11,704 Gg $CO_2$⁻ᵉ⁹ (MMA, 2016).

In order to estimate GHGs emissions bottom-up (based on fuel consumption and anthropogenic activity data) and top-down methods (based on atmospheric observations and modelling) are both widely applied and the scientific community is focusing on reducing their related uncertainties (e.g. Vermeulen et al., 2006; Bergamaschi et al., 2010; NRC, 2010; Jeong et al., 2013; Hiller et al., 2014). Top-down methods usually require high-quality and long-term GHGs observations. European projects, such as InGOS (www.ingos-infrastructure.eu), and infrastructures, such as ICOS (www.icos-infrastructure.eu), aim to offer atmospheric $CO_2$ and non-$CO_2$ GHGs data to better understand GHG fluxes in Europe and adjacent regions.

Nevertheless, in southern European regions, such as Spain, there is still a significant lack of high-quality atmospheric GHGs observations. The Catalan Institute of Climate Sciences (IC3) has been working since 2010 within the ClimaDat project at the creation of a network of remote stations for continuous measurements of GHGs, tracers and meteorological parameters (www.climadat.es). The IC3 network mainly aims to monitor and study the exchange of GHGs between the land surface and the lower atmosphere (troposphere) in different ecosystems, which are characterized by different biogenic and anthropogenic processes, under different synoptic conditions.

Besides GHGs concentrations, co-located observations of additional gases can provide us with useful tracers for source apportionment studies or to help better understand atmospheric processes (e.g. Zahorowski et al., 2004). Particularly the radioactive noble gas radon ($^{222}Rn$), due to its chemical and physical characteristic (e.g. Nazaroff and Nero, 1988), is being



extensively used for studying atmosphere dynamics, such as boundary layer evolution (e.g. Galmarini, 2006, Vinuesa and Galmarini, 2007), and soil-atmosphere exchanges (e.g. Schery et al., 1998; Zahorowski et al., 2004; Szegvary et al., 2009; Grossi et al., 2012; Vargas et al., 2015; Grossi et al., 2016). European GHGs monitoring infrastructures are already including
atmospheric $^{222}$Rn monitors in their stations (e.g. Arnold et al., 2010; Zimnoch et al., 2014; Schmithüsen et al., 2016). The co-evolution of atmospheric $^{222}$Rn and GHGs concentrations can also be used within the Radon Tracer Method (RTM) to estimate local/regional GHGs fluxes (e.g. Van der Laan et al., 2010; Levin et al. 2011; Vogel et al. 2012; Wada et al., 2013; Grossi et al., 2014).

In this study the new time series of atmospheric $CH_4$ concentrations measured at the IC3 station of Gredos and Iruelas (GIC3) between January 2013 and December 2015 has been analyzed.
The main aim was to investigate the major causes influencing the daily and seasonal variability of methane concentrations in a mountainous rural southern European region. The GIC3 station is located on the Spanish plateau, an area mainly characterized by livestock activity and where the transhumance is still practiced (Ruiz Perez and Valero Sáez, 1990). This is an ancestral activity consisting of the seasonal movement of the livestock over large distances to reach warmer regions during the winter and a return to the mountains in summer where pastures are more prosperous and suitable for grazing activities (Ruiz Perez and Valero Sáez, 1990; López Sáez et al., 2009). Particularly, the livestock leaves the GIC3 region to go to southern Spanish regions, such as Extremadura, during the cold period. The enteric fermentation due to digestive processes in animals can, thus, be a significant $CH_4$ source in this area.
The Unión de Pequeños Agricultores (UPA, 2009) reports that between 2004 and 2009 an average of 800,000 transhumant animals were hosted in Spain and 40,000 (5% of total) were counted in the province of Ávila, where GIC3 station is located. According to the available literature, in this area 85% of livestock still performs transhumance, with 500 stockbreeders moving every winter from the Gredos Natural Park (GNP) to warmer areas of Spain, such as Extremadura (Ruiz Perez and Valero Sáez, 1990; López Sáez et al., 2009; Libro Blanco, 2013). Generally, this mobility of the cattle and its associated $CH_4$ emissions (i.e. a major regional $CH_4$ source) cannot easily be included in country-wide (annual) inventories because it is not properly quantified and reported by nations. The present study wants to highlight the utility of $^{222}$Rn as tracer to retrieve independent GHGs fluxes on a monthly basis using atmospheric
$^{222}$Rn and $CH_4$ concentrations data. This work represents a first step toward a better further characterization of "mobile" sources, such as transhumant livestock for $CH_4$, which could help to improve national emissions inventories. Finally, it offers new $CH_4$ data for an under-sampled area which will help in the improvement of the regional and global methane budgets.

GIC3 is a new atmospheric station thus its location, the surrounding region and the instrumentation used at this station have been described in the methodology section of this manuscript. In the first part of the results section the daily and seasonal changes in $CH_4$ concentrations observed at the GIC3 station have been analysed in relation to $^{222}$Rn and PBLH variability. In
the second part, the local $CH_4$ fluxes and their monthly variability have been estimated by Radon Tracer Method (RTM), following Vogel et al. (2012), and using an emission inventory for $CH_4$ (EDGARv4.2). Both source estimation methods have been applied taking into account the same source region as modelled by the atmospheric transport model FLEXPARTv6.2. The possible influence of big cities surrounding GIC3 and of seasonally changing meteorological conditions on the retrieved $CH_4$ fluxes has also been investigated. Finally, the difference in $CH_4$ fluxes between the warm season, defined by the presence of the livestock in the GIC3 region, and the cold season, when the transhumant cattle migrates to the south of Spain, has been estimated.

## 2 Methods

### 2.1 Study site: Gredos and Iruelas station (GIC3)

The Gredos and Iruelas station (GIC3) is located in a rural region of the Spanish central plateau (40.35ºN; 5.17ºE; 1440 m above sea level (a.s.l.), Figure 1). GIC3 is set in the west side of the GPN, which has a total extension of 86,397 ha. The mountains of the GNP form the highest mountain range in the E-W orientated central mountain system that divides the Iberian
Peninsula in two parts. The GNP is located in a granitic basement; this type of soil presents high activity levels of $^{228}$U (Nazaroff and Nero, 1988). The average $^{222}$Rn flux in this area is of about 70-100 Bq m$^{-2}$ h$^{-1}$ (e.g. López-Coto et al., 2013; Karstens et al., 2015) which is almost twice the average radon flux in central Europe (Szegvary et al., 2009, López-Coto et al., 2013; Grossi et al., 2016). The vegetation at GIC3 area is stratified according to the altitude and the main land use practice is a mixture of agro-forestry exploitation (Figure 1).

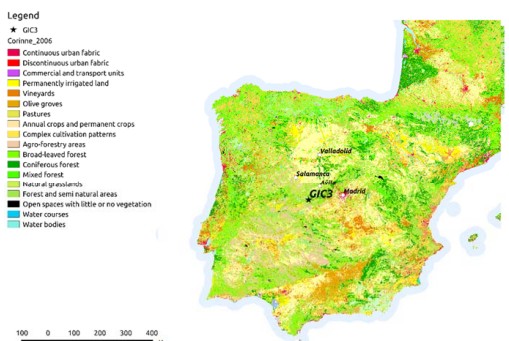

Legend
★ GIC3
Corinne_2006
  Continuous urban fabric
  Discontinuous urban fabric
  Commercial and transport units
  Permanently irrigated land
  Vineyards
  Olive groves
  Pastures
  Annual crops and permanent crops
  Complex cultivation patterns
  Agro-forestry areas
  Broad-leaved forest
  Coniferous forest
  Mixed forest
  Natural grasslands
  Forest and semi natural areas
  Open spaces with little or no vegetation
  Water courses
  Water bodies

Figure 1. CORINE land cover map 2006 for Spain with GIC3 (star label) and surrounding large cities (Madrid, Salamanca, Valladolid and Ávila).

Particularly, livestock farming is one of the main economic activities in the area around GIC3 station (Ruiz Perez and Valero Sáez, 1990; López Saéz et al., 2009; MMA, 2016; Hernández, 2016). In the GNP the seasonal migration of livestock starts in early November, when they travel to the south of the Iberian Peninsula, and they do not return until late May-mid June (Ruiz Perez and Valero Sáez, 1990). In Figure 2 a map of the main Spanish transhumant paths is presented. The path used by the livestock present at GIC3 region is presented as a zoom-in subplot,
indicating the entrance location (Puerto del Pico). Unfortunately, specific reports with data about the mobility rate of cattle or a local livestock count for individual months of the year are not available for the GIC3 area.

Besides livestock activities, there are four medium-size to large cities in the wider area surrounding GIC3. Several activities present in these cities, e.g. landfills or waste water treatment plants, represent $CH_4$ sources which could influence methane concentrations observed at GIC3 station under specific synoptic conditions. The metropolitan area of Madrid which comprises




about 6.3 million inhabitants is situated ca. 120 km to the east of GIC3. Valladolid, located 150 km to the west of GIC3, is reported to have ca. 416,000 inhabitants, while smaller cities like Salamanca (84 km to the north-west) and Ávila (55 km to the north-east) only have 229,000 and 59,000 inhabitants, respectively. More information about these four cities is reported in Table S1 of the supplement.

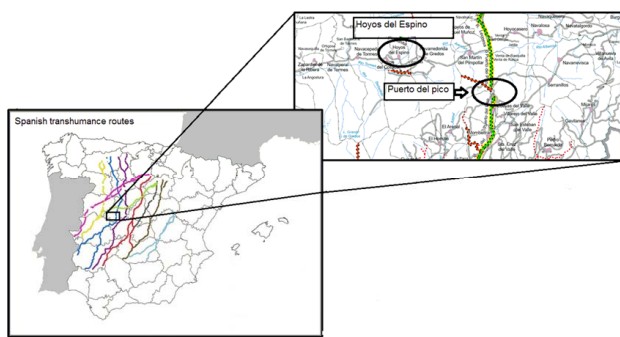

Figure 2. Spanish transhumance paths (left panel), Hoyo del Espino (location of IC3-GIC3 station) and Puerto del Pico (entrance to the GNP valley of the transhumant livestock coming in June from Extremadura, where they spent the winter months, across the Cañada Real Leonesa Occidental (green line of right panel)). Source: Ministerio de Agricultura, Alimentación y Medio Ambiente (Libro Blanco, 2013).

**2.2 Atmospheric measurements of CH₄ and ²²²Rn**


**2.2.1 Air sampling**

Atmospheric $CH_4$, $CO_2$ and $^{222}Rn$ concentrations are continuously measured since November 2012 at GIC3 station (air inlet at 20 m above ground level (a.g.l.) tower). $CH_4$ and $CO_2$ are measured using a G2301 analyzer (Picarro Inc., USA; Crosson, 2008) with a frequency of 0.2 Hz. Hourly atmospheric $^{222}Rn$ concentrations are measured using an Atmospheric Radon

MONitor (ARMON) (Grossi et al., 2012; Grossi et al., 2016).

The Picarro Inc. G2301 analyzer is calibrated every two weeks using 4 secondary working gas standards, which are calibrated at the beginning and at the end of their lifetime against seven standards of the National Oceanic and Atmospheric Administration (NOAA) (calibration scales are WMO-$CO_2$-X2007 and WMO-$CH_4$-X2004 for $CO_2$ and $CH_4$, respectively). A target gas is analyzed daily for 20 minutes to check the stability of the instrument. The instrument accuracy for $CH_4$ is of 0.36 ppb, calculated according to the definitions of the World Meteorological

Organization (WMO). The ARMON instrument was installed at the GIC3 station in collaboration with the Institute of Energetic Techniques of the Universitat Politècnica de Catalunya (INTE-UPC). The ARMON is a self-designed instrument based on α spectrometry of $^{218}Po$, collected electrostatically on a passivated implanted detector. The monitor has a minimum detectable activity of about 150 mBq m$^{-3}$ (Grossi et al., 2012). The performance of the ARMON has been previously tested against a widely used $^{222}Rn$ progeny monitor and good results have been observed (Grossi et al., 2016).

The responses of both ARMON and Picarro Inc. G2301 analyzers are influenced by the air sample humidity level. Water correction factors for both instruments are empirically determined and corrected following Grossi et al. (2012) and Rella (2010) methodologies, respectively.

**2.2.2 Drying system**

The instruments used at the GIC3 station require a total flow of 3 L min$^{-1}$ of sample air dried to a water concentration lower than 1000 ppm to perform simultaneous measurements of GHGs and $^{222}Rn$ concentrations. In the GIC3 inlet system the sample air is passed through a Nafion® membrane (Permapure, PD-100T-24MPS) that exchanges water molecules with a dry counter-current. The counter-current air flow is dried in a two steps process, first through a cooling coil in a refrigerator at 3 ºC and a pressure of 5.5 barg, and then using a cryotrap at -70 ºC at a pressure of 1.5 barg. Multiple cryotraps are selected with electrovalves in order to increase the autonomy of the system to about 2 months. The typical water content of sample air inside the instruments is between 100 and 200 ppm.


**2.2.3 Meteorological observations**

Meteorological variables are continuously measured at the GIC3 tower. The canopy around the tower is below 20 cm and the surrounding area is quite hilly. The tower is equipped with: (1) Two-dimensional sonic anemometer (WindSonic, Gill Instruments) for wind speed and direction (accuracies of ± 2 % and ± 3 º, respectively); (2) Humidity and temperature probe (HMP

110, Vaisala) with an accuracy of ± 1.7 % and ± 0.2 ºC, respectively; (3) Barometric pressure sensor (61302V, Young Company) with an accuracy of 0.2 hPa (at 25 °C) and 0.3 hPa (from - 40 to +60 °C). All the accuracies refer to manufacturer's specifications.

**2.3 Planetary boundary layer height (PBLH)**

Planetary boundary layer height (PBLH) data used in this analysis have been extracted from the operational high resolution atmospheric model of the European Center for Medium-Range Weather Forecasting (ECMWF-HRES) (ECMWF, 2006) for the period of interest (January 2013 - December 2015) at GIC3 area. This model stores output variables every 12 hours (at 00.00 UTC and 12.00 UTC) with a temporal resolution output of 1 h and with forecasts from +00h to +11h. The horizontal spatial resolution of the model is about 16 km. In the ECMWF-HRES model the calculation of the PBLH is based on the bulk Richardson number (Ri) (Troen and Mahrt, 1986).




### 2.4 CH₄ fluxes

#### 2.4.1 FLEXPART_RTM_CH₄ fluxes (FR_CH4)

The RTM is a well known method (e.g. Hammer and Levin 2009) and it has been used in this study, following the implementation described in Vogel et al. (2012), to obtain observation-based estimates of the nocturnal $CH_4$ fluxes at GIC3. The RTM uses atmospheric measurements of $^{222}Rn$ and measured, or modelled, values of its $^{222}Rn$ fluxes together with atmospheric concentrations of a gas, i.e. $CH_4$, in order to retrieve the net fluxes of the latter gas (e.g. Hammer and Levin 2009; Grossi et al., 2014). This method is based on the main assumption that the nocturnal lower atmospheric boundary layer can be described as a well-mixed box of air (Schmidt et al. 1996; Levin et al., 2011; Vogel et al., 2012). In this atmospheric volume the variation of the concentration of any tracer with time $C_i(t)$ will be proportional to the flux of the tracer $F_i(t)$ and inversely proportional to the height of the boundary layer $(h_i(t))$ (Eq.1; e.g. Galmarini, 2006; Griffiths et al., 2012; Grossi et al., 2014).

$$\frac{dC_i(t)}{dt} \propto F_i(t) \cdot \frac{1}{h_i(t)} \qquad (1)$$

The boundary layer is considered homogeneous within the box and with a time varying height. No significant horizontal advection is considered due to stable atmospheric conditions (Griffiths et al., 2012). Observing the concentration increase of two gases that fulfil the assumptions, here $CH_4$ and $^{222}Rn$, and knowing the flux of $^{222}Rn$ then the flux of $CH_4$ can be calculated (Levin et al., 2011). A description of the specific criteria used to implement the RTM, which include selection criteria to reject situations with unstable atmospheric conditions, remote influences on the concentration and outliers detection, can be found in detail in Vogel et al. (2012). Grossi et al. (2014) previously applied the RTM for the first time at the GIC3 station using only a 3-months dataset and with a constant (in time and space) $^{222}Rn$ flux of 60 Bq m$^{-2}$ h$^{-1}$. Here, in order to apply the RTM to retrieve a time series of $CH_4$ fluxes (FR_CH4) during 2013-2015 at the GIC3 station and to compare these results with the ones obtained using a bottom-up inventory for methane (FE_CH4), we used the following extensive setup:

1. A nocturnal window between 20.00 UTC and 05.00 UTC was selected for the analysis to utilize only accumulation events when atmospheric concentrations of both $CH_4$ and $^{222}Rn$ had a positive concentration gradient due to positive net fluxes under stable boundary layer conditions;

2. A data selection criterion based on a threshold of $R^2 \geq 0.8$ for the linear correlation between $^{222}Rn$ and $CH_4$ was used to reject events with low linear correlation between the atmospheric concentrations of both gases;

3. An *effective* local radon flux influencing GIC3 station each night from 2013 to 2015 was calculated coupling radon flux data, obtained using a model developed by López-Coto et al. (2013), with the footprints calculated by ECMWF-FLEXPART model (version 6.02) (Stohl, 1998). Radon flux data were calculated as explained in the following paragraph and the footprints obtained are described in section 2.4.3.

.

The radon flux model (from now on named UHU model) employed in this work has been described in detail by López-Coto et al. (2013). By using this model, a time-dependent inventory was calculated for the period 2011–2014 employing several dynamic inputs, namely soil moisture, soil temperature and snow cover thickness. These data were obtained directly from Weather Research and Forecasting (WRF) simulations (Skamarock et al., 2008). A domain of 97 x 97 grid cells centred in Spain with a spatial resolution of 27 x 27 km$^2$ and a temporal resolution of 1 hour was defined. $^{222}Rn$ flux data calculated using this model were only available until November 2014 due to a lack of WRF simulations. In order to fulfil the period when modelled $^{222}Rn$ flux data were not available, from December 2014 to December 2015, a monthly climatology was calculated using the data set of UHU model for the years 2011-2014.

### 2.4.2 FLEXPART_EDGAR_CH₄ fluxes (FE_CH4)

Bottom-up $CH_4$ fluxes influencing GIC3 station were estimated using the footprints calculated by ECMWF-FLEXPART model (obtained as described in section 2.4.3) and the Emissions Database for Global Atmospheric Research (EDGAR) version 4.2 (EDGAR, 2010). EDGAR inventory, developed by the European Commission Joint Research Centre and the Netherlands Environmental Assessment Agency, includes global anthropogenic emissions of GHGs and air pollutants by country and on a spatial grid. The EDGAR version used in the present study provides spatial (cells of 0.1 degree) annual mean $CH_4$ emissions globally. All major anthropogenic source sectors, e.g. waste treatment, industrial and agricultural sources (e.g. enteric fermentation), are included, whereas natural sources (e.g. wetlands or rivers) are not.

The influence of the emissions associated to the cities surrounding the region of GIC3 was also modelled to better understand their impact. In Table S1 of the supplement the coordinates of the upper and lower corners of the areas used to describe the location of the metropolitan areas over the EDGAR inventory are reported.

### 2.4.3 Footprints

The lagrangian particle dispersion model FLEXPARTv6.2 has been extensively validated and is nowadays widely used by the scientific community to calculate atmospheric source-receptor relationships for atmospheric gases and organic particles (e.g. Stohl, 1998; Stohl et al., 2005; Arnold et al., 2010; Font et al., 2013; Tohjima et al., 2014). FLEXPART allows the computation of the trajectories of virtual air parcels arriving at the receptor point, i.e. the GIC3 station, at a specific time. FLEXPART has been applied here to calculate 24 h backward trajectories of 10,000 virtual air parcels starting at 00.00 UTC for each night of the period 2013-2015. Each back trajectory simulation was run with a time-step output of 3 h. Meteorological data from the operational ECMWF-HRES model with a resolution of 0.2 degrees were used as input fields for the FLEXPART modelling. The FLEXPART output domain resolution was of 0.2 degrees. The domain was set at (25ºN, 40ºW) for the lowest left corner and (65ºN, 10ºW) for the upper right corner. A nested output domain of 0.05 degrees resolution was defined at (37ºN, 12ºW) for the lowest left corner and (43ºN, 0ºE) for the upper right corner. The FLEXPART model accounts for the vertical and horizontal position of the virtual air parcels and their residence time in each grid cell. This information allows estimating the influence of the atmosphere-surface exchange on the observed concentrations if air parcels are within the boundary layer. A maximum height of 300 m a.g.l. has been selected for the footprint analysis following Font et al. (2013).

The footprints obtained for the nested FLEXPART domain were combined with the EDGAR inventory map for $CH_4$ emissions (EDGAR, 2010) and with the UHU $^{222}Rn$ flux inventory map (López-Coto et al., 2013), separately, in order to obtain the time series of modelled $CH_4$ and $^{222}Rn$ fluxes. The resulting mean flux $F_C(S,T)$, for each gas C, at the receptor S and at time T is thus given by Eq. 2:






$$F_C(S, T) = \sum_{t=t_o}^{t=T} \sum_x F_C(x, t) * w(x, t) \qquad (2)$$

where $F_C(x,t)$ denotes the flux of a given grid cell x at time t derived from the EDGAR or UHU inventory map, separately. The weighting factor of each grid cell $w(x,t)$ is calculated using
FLEXPART footprint for each night over the 3 years period and it has been calculated normalizing the residence time of each grid cell over the nested domain.


### 3 Results

In the present section we present the results of the daily and seasonal atmospheric $CH_4$ variability at GIC3 station analysed using a record of 3-years hourly $CH_4$ and $^{222}Rn$ time series.
Unfortunately, due to problems in the air sample system the 11 % of the total data set was not available.


Since Grossi et al. (2016) presented a complete characterization of the main meteorological conditions and $^{222}Rn$ behaviour at GIC3, along with other Spanish stations, we will use these
previous results to interpret the variability of the atmospheric processes and the variability of $CH_4$ concentrations, as well as to interpret the dominating wind regimes for $CH_4$ flux data
analysis (Figures S1 and S2 of the supplement present the daily and monthly $^{222}Rn$ variations and the monthly wind regimes observed at GIC3 station).

### 3.1. Statistics of the daily and seasonal atmospheric $CH_4$ variability

The 3-years hourly time series of atmospheric $CH_4$ concentrations measured at the rural area of GIC3 shows a median value over the dataset is 1904.5 ppb with an absolute deviation of
29.6 ppb. The boxplots in Figure 3 present the medians of the atmospheric $CH_4$ concentrations measured at GIC3 station over the dataset on an hourly (left panel) and a monthly (right
panel) basis.

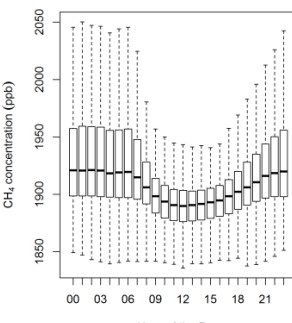
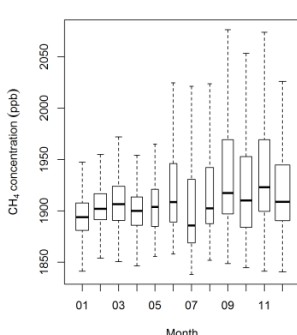


Figure 3. Boxplots of hourly (left panel) and monthly (right panel) atmospheric $CH_4$ concentrations measured from January 2013 to December 2015 at GIC3 station. For each median (black
bold line) the 25th (Q1; lower box limit) and 75th (Q3; upper box limit) percentiles are reported in the plot. The lower whisker goes from Q1 to the smallest non-outlier in the data set, and
the upper whisker goes from Q3 to the largest non-outlier. Outliers are defined as >1.5 IQR or <1.5 IQR (IQR: Interquartile Range).

The maximum hourly median methane concentration measured within the 3 years of observations is 1921.1 ppb and is observed at 03.00 UTC, whereas the minimum median value of
1889.9 ppb is observed at 13.00 UTC. The absolute standard deviation of the median is 16.97 ppb. The median daily amplitude at this station, between the minimum and the maximum, is
of 31.18 ppb. $CH_4$ concentrations usually start decreasing at GIC3 in the morning around 07.00 UTC and 08.00 UTC and begin to increase again in the afternoon around 17.00 UTC and
18.00 UTC. Nighttime $CH_4$ concentrations present an absolute standard deviation of 60 ppb while for daytime concentrations it is of 30 ppb. For the monthly medians, Figure 3 (right panel)
shows that atmospheric median methane concentrations range between 1885.8 ppb and 1923.1 ppb. A light increase of methane concentrations seems to be observed between the first and
the second semester of the year.

### 3.2 Daily and seasonal PBLH variability

Figure 4 shows the daily and seasonal variability of the PBLH at GIC3 station. It can be observed that on a daily basis the PBLH reaches its minimum between 01.00 UTC and 07.00 UTC.
Indeed, within this interval median PBLH values present minima of 45 m a.g.l. The PBLH starts to increase around 08.00 UTC, reaching its maximum between 14.00 UTC and 16.00 UTC
and then decreases again after 17.00 UTC. The maximum median PBLH value is 1037 m a.g.l. The absolute standard deviation is 283 m a.g.l.. On a monthly basis, the median PBLH
reaches its minimum during winter months, January and December, with a value of 204 m a.g.l. The highest PBL heights are observed in summer months with typical values around 595 m
a.g.l. and an absolute standard deviation of 204 m a.g.l.. The monthly PBLH is quite symmetric (around July as center-line) and many months in fall and spring experience similar PBLH
distributions.




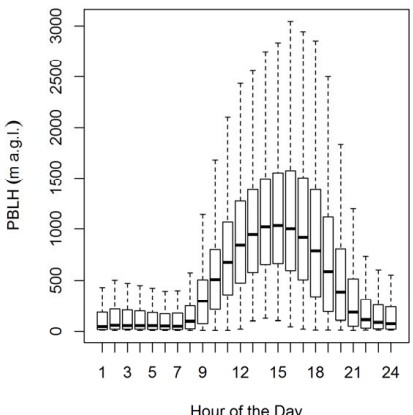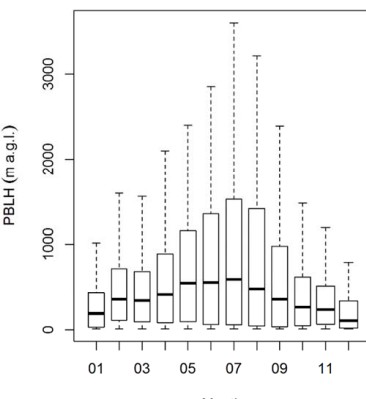

Figure 4 Boxplots of hourly (left panel) and monthly (right panel) PBLH data extracted from ECMWF-HRES model for the period January 2013 - December 2015 at GIC3 station. For each median (black bold line) the 25th (Q1; lower box limit) and 75th (Q3; upper box limit) percentiles are reported in the plot. The lower whisker goes from Q1 to the smallest non-outlier in the data set, and the upper whisker goes from Q3 to the largest non-outlier. Outliers are defined as >1.5 IQR or <1.5 IQR (IQR: Interquartile Range).

### 3.3 Comparison between $CH_4$ and $^{222}Rn$ variability


A comparison of the daily and seasonal variability of the atmospheric concentrations of $^{222}Rn$ and $CH_4$ in relation to the changes in the height of the PBL at GIC3 station (2013-2015) is presented in Figures 5 and 6, respectively.

The daily evolution of hourly $^{222}Rn$ atmospheric concentrations (Figure 5, upper panel) implies that on daily time-scale, when $^{222}Rn$ flux can be considered fairly constant (e.g. López-Coto
et al., 2013), PBLH variations drive the increase or decrease of the atmospheric $^{222}Rn$ concentrations. In this sense, $^{222}Rn$ seems to be an excellent predictor of PBLH (and vice versa) on a daily time-scale. Looking at the hourly means of the atmospheric $CH_4$ concentrations (Figure 5, lower panel) we can observe that the daily methane evolution also decreases in agreement with the increase of the PBLH, as it was observed for $^{222}Rn$. However, between 16.00 UTC and 18.00 UTC higher values in $CH_4$ concentrations relative to the values observed during 10.00 UTC and 12.00 UTC are observed, which have similar PBLH conditions which could indicate some daily variability in the methane fluxes.





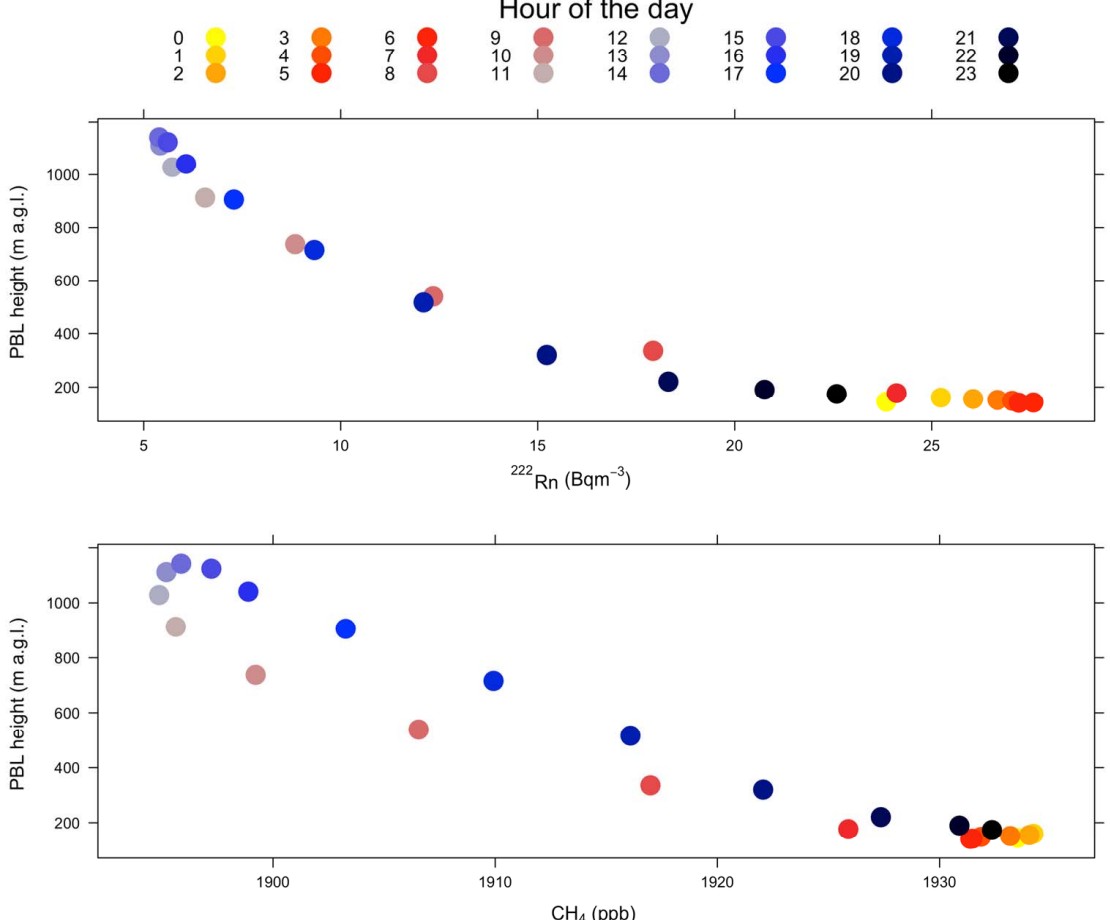

Figure 5 Relation between hourly means of atmospheric CH$_4$ (lower panel) and $^{222}$Rn (upper panel) concentrations measured during 2013-2015 at GIC3 station and ECMWF data of PBLH
at the same area and during the same time interval.

To interpret the monthly variability, the daily amplitude for each gas, i.e. Δ$^{222}$Rn$_{daily}$ for radon and ΔCH$_{4daily}$ for methane, was calculated in order to subtract the influence of the changing daily background contribution measured at GIC3 station. Then, Δ$^{222}$Rn is defined as the difference between average nighttime concentration data (18.00 UTC - 07.00 UTC) versus average daytime (08.00 UTC-17.00 UTC) concentrations data (Eq. 3). ΔCH$_4$ has been calculated accordingly.


$$\Delta\,^{222}Rn = \left\langle\,^{222}Rn_{\,nighttime}\,\right\rangle - \left\langle\,^{222}Rn_{\,daytime}\,\right\rangle \qquad (3)$$

Figure 6 reveals that monthly amplitudes increase in summer, when the daytime PBLH increase very strongly due to vertical mixing (see Figure 4). This general tendency is found both for $^{222}$Rn and CH$_4$ concentrations. $^{222}$Rn concentrations amplitudes in autumn are higher than in winter under the same PBLH conditions (Figure 6, upper panel). This could indicate that some
process, other than PBLH, is driving this difference of the $^{222}$Rn concentrations. In Figure 7, it can be observed how the monthly $^{222}$Rn flux calculated by the UHU model (presented in section 2.4) changes, where the circles indicating each month have been coloured in agreement with Figure 6.

In agreement with the results discussed by Grossi et al. (2016), the $^{222}$Rn flux at GIC3 is lower during winter, due to snow cover events and low temperatures which prevent $^{222}$Rn diffusion from the soil. Then, it increases almost two-fold and three-fold during the autumn and summer months, respectively. This is due to drier soil conditions and the high gradient of temperature
in the surface atmospheric layer which facilitates $^{222}$Rn to escape from the pores of the granitic soil (Nazaroff and Nero, 1988). This seasonality of the $^{222}$Rn flux could be the main cause of the increased atmospheric Δ$^{222}$Rn under the same PBLH conditions.

Monthly variations of ΔCH$_4$ shown in Figure 6 (bottom panel) also display no clear simple correlation with PBLH. Indeed, ΔCH$_4$ appears higher between the months of June and December
independently from the corresponding PBLH values.



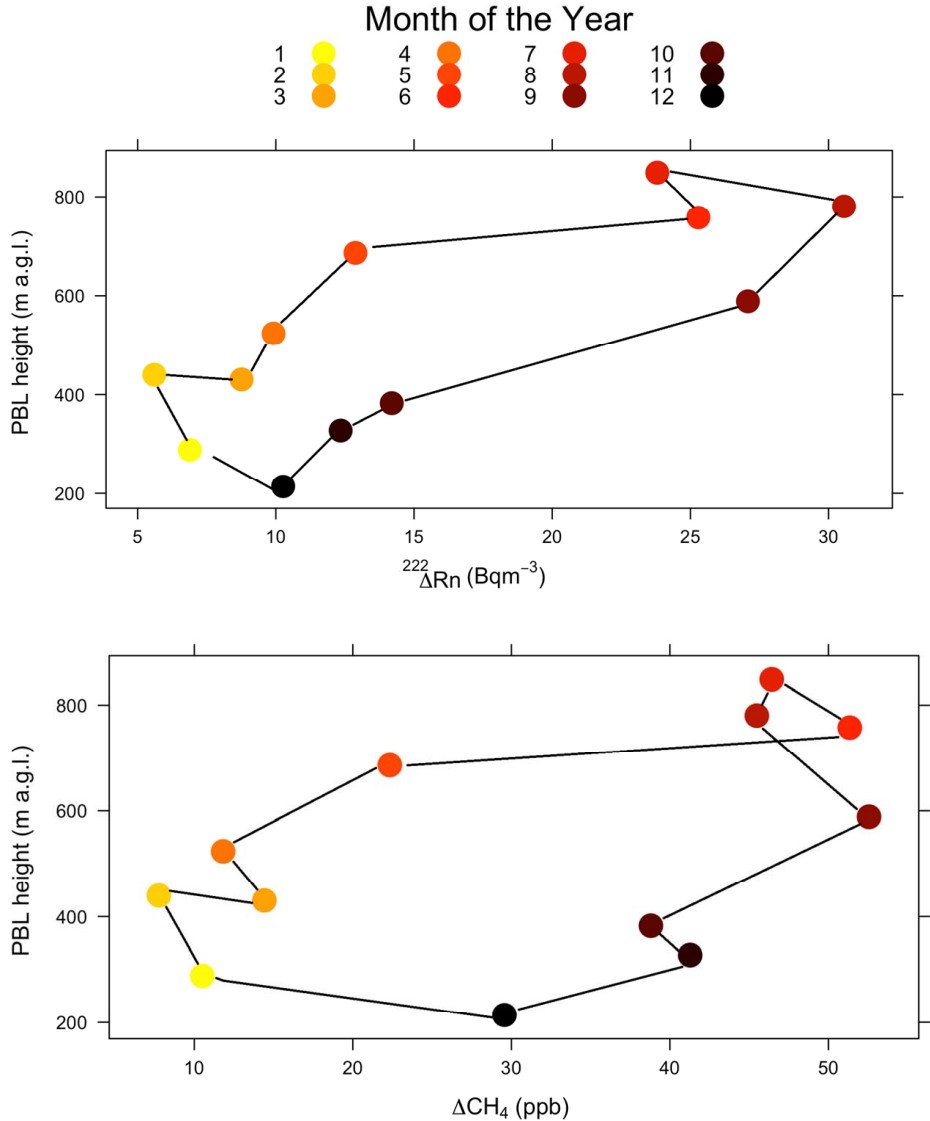


Figure 6. Relation between monthly means of concentration amplitudes of ΔCH₄ (bottom panel) and Δ²²²Rn (upper panel) measured during 2013-2015 at GIC3 station and monthly ECMWF data of PBLH at the same area during same time interval.



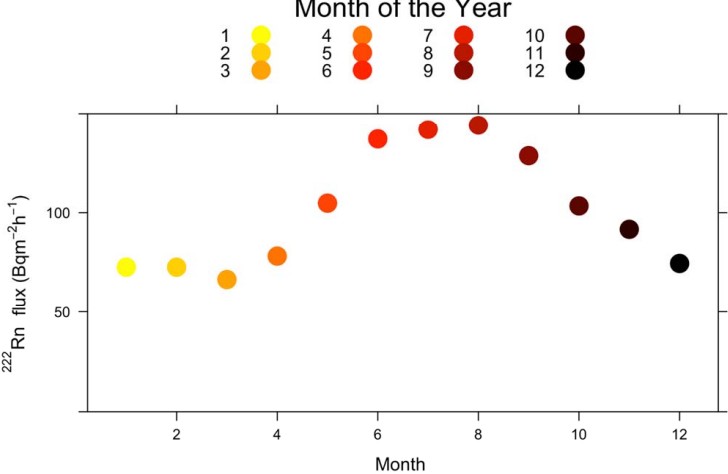

315                             Figure 7. Monthly $^{222}$Rn flux means calculated by the UHU model and climatology for 2013-2015 at the GIC3 station.

### 3.4 Variations of CH₄ fluxes

So far daily variations for both CH₄ and $^{222}$Rn concentrations can be explained in relation to the accumulation or dilution of gas concentration within the PBL. Monthly Δ$^{222}$Rn variability
can be understood when we account for seasonal $^{222}$Rn flux changes. Unfortunately, existing emission inventories (EDGAR, 2010; MMA, 2016) do generally not yet provide seasonally
and hourly varying CH₄ emission values for Europe in general nor for Spain in particular.

In order to understand the impact that temporal changes of CH₄ emissions may have on monthly mean atmospheric CH₄ concentrations we have applied two different methodologies, as
explained earlier, and we compared their resulting fluxes: FR_CH₄ and FE_CH₄, respectively. Figure 8 presents the *effective* $^{222}$Rn flux time series used for the application of the first
methodology (RTM), together with the raw $^{222}$Rn flux calculated by the UHU model and its climatology.

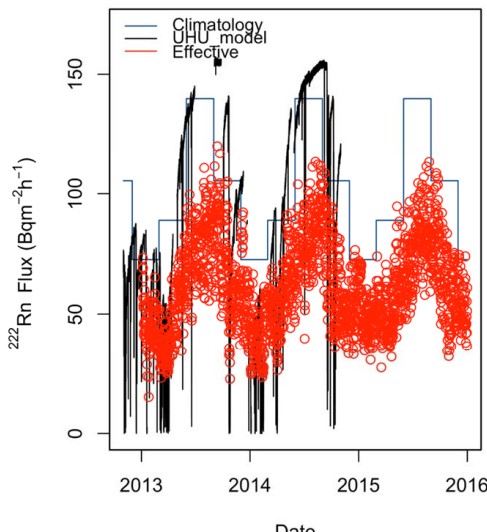

Figure 8 Time series of local $^{222}$Rn flux calculated by UHU model (black line; López-Coto et al. (2013)) for GIC3 area, $^{222}$Rn flux climatology (blue line) and *effective* $^{222}$Rn flux calculated
on the basis of FLEXPART footprints (red dots). This last series was used within the RTM method.



Figure 9 presents the time series of CH₄ fluxes estimated at GIC3 station and $T_i$ (grey shaded rectangles) indicates the time when transhumant livestock returns to the GNP after spending

the winter in the south of Spain (Tapias, 2014; Rodríguez, 2015). The green shaded areas indicate the periods, between June and December, when transhumant livestock typically stays in the GIC3 region (Ruiz Perez and Valero Sáez, 1990; López Sáez et al., 2009; Libro Blanco, 2013). The mean FR_CH₄ flux is of 0.19 mg CH₄ m⁻² h⁻¹ with 25th and 75th percentiles of 0.11 mg CH₄ m⁻² h⁻¹ and 0.23 mg CH₄ m⁻² h⁻¹, respectively. Data coverage in the second part of the time-series (2014-2015) is significantly higher than in the first period (2013-2014) because the simultaneous availability of ²²²Rn and CH₄ data was higher. FE_CH₄ fluxes are higher, with an annual mean value of 0.33 mg CH₄ m⁻² h⁻¹ and with 25th and 75th percentiles of 0.28 mg CH₄ m⁻² h⁻¹ and 0.36 mg CH₄ m⁻² h⁻¹, respectively. Furthermore, FEC_CH₄ fluxes obtained with the EDGARv4.2 inventory and considering only the contribution of the cities that are located

around GIC3 station, in agreement with the masks presented in Table S1 of the supplement material, had a mean value of 0.02 mg CH₄ m⁻² h⁻¹ with 25th and 75th percentiles of 0 mg CH₄ m⁻² h⁻¹ and 0.01 mg CH₄ m⁻² h⁻¹. respectively.

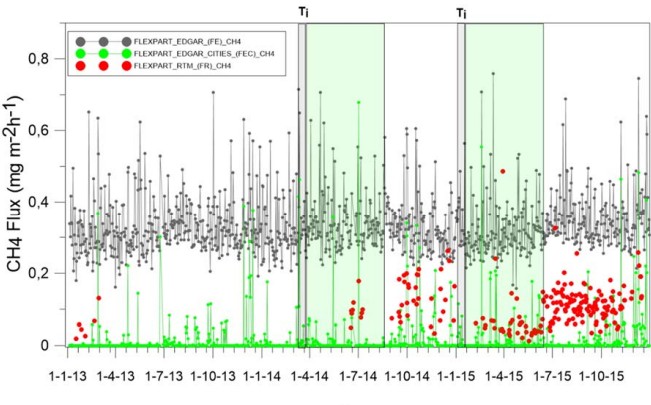

Figure 9  Results of nighttime FR_CH₄ fluxes (mg CH₄ m⁻² h⁻¹) (red circles) obtained at GIC3 station from January 2013 to December 2015 compared with nighttime FE_CH₄ fluxes obtained using bottom-up inventory emissions (grey line) and calculated FEC_CH₄ fluxes surrounding cities contributions (green circles).The weeks $T_i$ represent the period of 2014 (21st-27th June) and 2015 (20th-26th June), concurrent with the availability of FR_CH₄ fluxes data, when transhumant livestock came back to GIC3 area after spending the winter in the south of Spain. Shaded green regions represent the orientative periods when transhumant livestock stay in GIC3 area.

Figure 10 shows monthly boxplots of FE_CH₄ and FR_CH₄ fluxes. Shaded areas are coloured according to the main local wind directions arriving at GIC3 station. This classification is based on the results presented in Figure S2 of the supplementary material, where monthly windrose plots for GIC3 station between 2013-2015 are shown. We can observe that there is no significant variability in monthly FE_CH₄ flux values. In contrast, FR_CH₄ flux results show a noticeable increase of CH₄ fluxes between June and December that seems to be independent of the seasonally changing dominant wind directions. This is also uncorrelated with seasonally changing ²²²Rn fluxes (Figure 7).

The seasonal change of CH₄ fluxes between the first and the second half of the year at GIC3 could be indeed related to variations in the local CH₄ emissions. In addition, the highest FR_CH₄ flux values were observed in December, which also coincides with an increase of winds coming from the east in agreement with the monthly methane flux, based on EDGARv4.2, from the cities (FEC_CH₄). Overall, cities contribution is only visible during certain months, especially when dominant wind conditions come from the East in the direction of the Madrid urban area (see Figure S2 of the supplement material). During the second semester of the year the difference between FR_CH₄ and FE_CH₄ fluxes is significantly reduced.

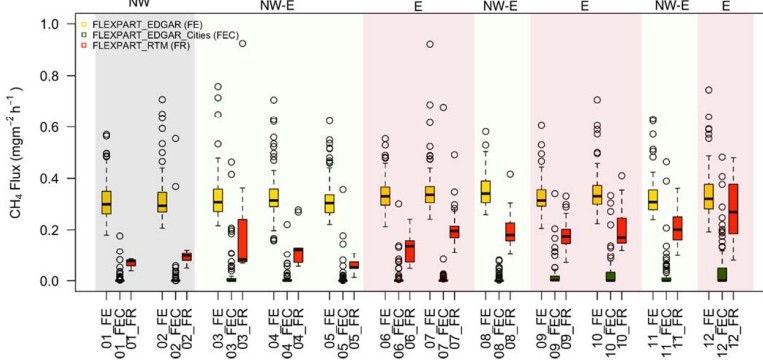

Figure 10 Boxplots of monthly CH₄ fluxes (mg CH₄ m⁻² h⁻¹) calculated at GIC3 area using the RTM technique (red) and the EDGAR inventory (total in yellow; cities contribution in green). Coloured areas indicate main wind directions for specific months. For each median (black bold line) the 25th (Q1; lower box limit) and 75th (Q3; upper box limit) percentiles are reported in the plot. The lower whisker goes from Q1 to the smallest non-outlier in the data set, and the upper whisker goes from Q3 to the largest non-outlier. Outliers are defined as >1.5 IQR or <1.5 IQR (IQR: Interquartile Range).



Finally, Figure 11 shows the boxplot of FE_CH$_4$ and FR_CH$_4$ fluxes aggregated according to the "cold" season, when there is no livestock in the GIC3 area, and "warm" season, when the animals are back to the valley. According to these data during January-May (cold season) FR_CH$_4$ fluxes present a median value of 0.08 mg CH$_4$ m$^{-2}$ h$^{-1}$ with an absolute deviation of 0.05 mg CH$_4$ m$^{-2}$ h$^{-1}$ and a median value of 0. 19 mg CH$_4$ m$^{-2}$ h$^{-1}$ with an absolute deviation of 0.06 mg CH$_4$ m$^{-2}$ h$^{-1}$ during June-December (warm season).

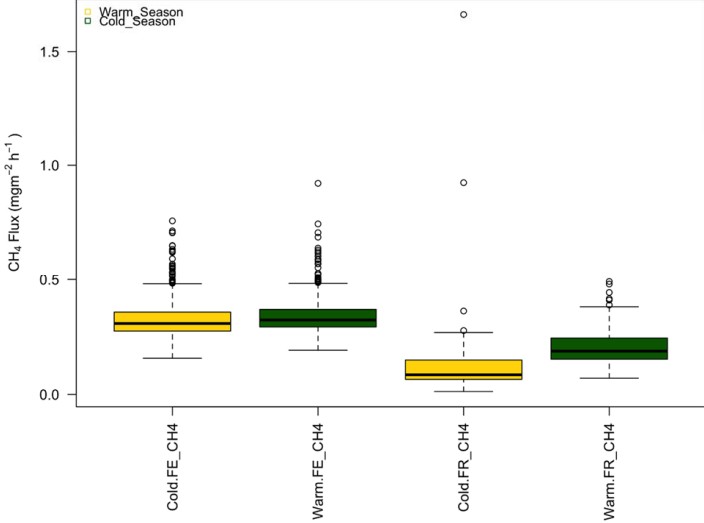


Figure 11 Boxplots of FE_CH$_4$ and FR_CH$_4$ fluxes (both in mg m$^{-2}$ h$^{-1}$) calculated at GIC3 area during the "warm" season (June-December, dark green box) and the "cold" season (January-May, yellow box). For each median (black bold line) the 25$^{th}$ (Q1; lower box limit) and 75$^{th}$ (Q3; upper box limit) percentiles are reported in the plot. The lower whisker goes from Q1 to the smallest non-outlier in the data set, and the upper whisker goes from Q3 to the largest non-outlier. Outliers are defined as >1.5 IQR or <1.5 IQR (IQR: Interquartile Range).

**4 Discussion**

The present results show the different influences that meteorological conditions (PBLH and wind direction) and regional fluxes have on the variability of atmospheric CH$_4$ concentrations observed at GIC3. $^{222}$Rn observations have been used, together with PBLH, to better understand the reasons of the variability of the atmospheric CH$_4$ concentrations observed at GIC3. Particularly, the power of the $^{222}$Rn as tracer to calculate independent fluxes of GHGs has been shown in order to help with the improvement of emission inventories on regional scale.


**4.1 Daily variability of atmospheric CH$_4$ concentrations**

The daily cycle of atmospheric CH$_4$ concentrations measured at GIC3 shows a significant variation between daytime and nighttime periods. The large increase of nocturnal CH$_4$ concentrations can be explained by the significantly decreased height of the planetary boundary layer (Figure 4), which is supported by a similar behaviour of $^{222}$Rn concentrations. Indeed,
CH$_4$, as well as $^{222}$Rn, reach their maximum concentration values during the night when the PBLH is under 200 m a.g.l. and their atmospheric concentrations decrease with the increase of the PBLH during daytime.

The good correlation of PBLH and $^{222}$Rn (and CH$_4$) in Figure 5 indicates that $^{222}$Rn fluxes do not strongly vary on daily time-scales or at least not to a degree that significantly influences their atmospheric concentration variability. Daily CH$_4$ fluxes seem to change on daily time-scale. Average afternoon CH$_4$ concentrations are slightly enhanced compared to those from the
morning for similar PBLH values (Figure 5, bottom panel). They show a small hysteresis behaviour which could indicate that local emissions slightly increase then, or that a systematic transport of CH$_4$ enhanced air-masses occur at GIC3 during the afternoon.

**4.2 Seasonal variability of atmospheric CH$_4$ concentrations**

To understand the impact of monthly changing PBLH, local meteorology and regional emissions, the interpretation of monthly data needs to account also for the changing background concentrations of CH$_4$ at GIC3. To take this issue into account, we discuss the mean monthly local enhancement of CH$_4$ ($\Delta$CH$_4$) between daytime and nighttime. This definition of $\Delta$CH$_4$ allows us to subtract seasonal and synoptic background variations, and focus on the impact of PBLH for individual days that are then averaged to investigate how $\Delta$CH$_4$ changes on a monthly basis. The observed variability of $\Delta$CH$_4$ (Figure 6, lower panel) cannot be explained only in terms of the changes of the PBLH. Monthly averages of $\Delta$CH$_4$ (and monthly CH$_4$ boxplots, Figure 3) present their maximum values between June and December; and their minimum values during the rest of the months independently of the height of the PBL.


From co-located $^{222}$Rn concentration observations we learn that a significant increase in the regional fluxes (Figure 7) can compensate the effect of increased dilution in the deeper summer PBL on the observed concentrations (Figure 6, upper panel). The increase of $^{222}$Rn flux in the GIC3 region from winter to autumn season and the following decrease can coherently explain the variation observed in monthly $\Delta^{222}$Rn. The comparison between $\Delta$CH$_4$ and $\Delta^{222}$Rn suggests that there may be also a strongly varying seasonal source of CH$_4$ which has been confirmed by our FR_CH$_4$ fluxes estimates, as seen in Figures 9, 10 and 11. Of course, the FR_CH$_4$ fluxes estimates are limited to nighttime, but, as previously discussed in section 4.1, we can assume





that the daily fluxes of $CH_4$ only vary to a small degree and we thus consider that the nocturnal RTM results are representative for the overall daily $CH_4$ fluxes. FR_$CH_4$ fluxes show a mean value 0.14 mg $CH_4$ m$^{-2}$ h$^{-1}$ lower than FE_$CH_4$ fluxes over the data set.

FR_$CH_4$ fluxes show an increase during the second semester of the year of 0.11 mg $CH_4$ m$^{-2}$ h$^{-1}$ on monthly basis. The increase seems to coincide with the period of the year when transhumant livestock resides in the GIC3 region. UPA (2009) reports that around 40,000 animals, mainly bovine, crossed the Puerto del Pico border of the Sierra de Gredos in June 2014 and June 2015

coming back after the winter. During this period of enhanced ruminant emissions, FR_$CH_4$ and FE_$CH_4$ fluxes are much more in agreement. This last result could be due to the constant emission factor of $CH_4$ emission used within the bottom-up inventory which, of course, cannot yet reflect transhumance activity. The RTM analysis performed here allows to observe the additional contribution to the regional $CH_4$ emissions due to livestock activity in the GIC3 area, which appears to be a dominant source in the second half of the year.

### 5 Conclusions and outlook


To gain a full picture of the Spanish (and European) GHGs balance the understanding of $CH_4$ emissions in the currently understudied regions is a critical challenge as well as the improvement of bottom-up inventories for all European regions. Our study uses, among others, GHGs, meteorological and $^{222}Rn$ tracer data from one of the eight stations of the new ClimaDat network which provides continuous atmospheric observations of $CH_4$ and $^{222}Rn$ in a systematically under-sampled region of Spain. The present study underlines that this data, combined with retrieved PBLHs data, atmospheric transport modelling (FLEXPARTv6.2) and a bottom-up emission inventory (EDGARv4.2), allows addressing the main causes of the spatial and temporal

variability of the regional GHGs sources and offer new tools to improve regional emission inventories related with temporal and/or moving human activities such as livestock.

Although no precise data on transhumant activity in Spain is available so far, our study highlights the importance of transhumance, as an anthropogenic activity for livestock management, in the regional $CH_4$ budget of central Spain. Establishing a clear link between regional $CH_4$ fluxes and the transhumance activity will allow accounting for this effect in future emission inventories of the region (and Europe). In addition, our results show that urban emissions can be transported and influence the atmospheric composition of remote rural areas over several

hundred kilometres under specific synoptic conditions.

Besides supporting better future temporal information for inventories, our findings could also be applied to monitor the impact of emission mitigation strategies on regional emission trends applying the RTM separately. Indeed, other researchers suggested that Best Management Practices (BMP) for cattle can drive a reduction of 22-30% in $CH_4$ emissions compared with continuous grazing management (De Ramus et al., 2003; FAO 2013). In this sense, the methodology applied in this study could be useful in the future to observe the impact of BMP on the

reduction of ruminant $CH_4$ emissions on a regional scale.

These first promising results motivate the further application of this RTM to other GHGs time series from the ClimaDat network, as well as in continent-wide networks such as ICOS that perform co-located GHGs and $^{222}Rn$ observations.


### Acknowledgements

The research leading to these results has received funding from "la Caixa" Foundation.with the ClimaDat project (2010-2014) and from the Ministerio Español de Economía y Competividad, Retos 2013 ( 2014-2016) with the MIP (Methane interchange between soil and air over the Iberian Península) project (reference: CGL2013-46186-R). This study was also possible thank

to the collaboration with the autonomous community of Castile and León and the Sierra de Gredos Regional Park.

C. G. particularly thanks the Ministerio Español de Educación, Cultura y Deporte to partially support her work with the research mobility grant "José Castillejos" (ref. CAs15/00042). The research of F.R.V. is funded and supported by the Chaire industrielle BridGES, a joint research program between ThalesAleniaSpace, Veolia and the parent institutions of LSCE (CEA, CNRS, UVSQ).


Authors warmly thank: i) the LAO (Atmosphere and Ocean laboratory) team, in the persons of: Rosa Arias, Manel Nofuentes, Oscar Batet, Lidia Cañas, Silvia Borras, Paola Occhipinti and Eusebi Vazquez, for their efforts in the installation, maintenance and calibration of all IC3 climatic stations and data, including the GIC3 where this study has been performed; ii) the INTE team, in the persons of: Vicente Blasco and Juan Antonio Romero, for their work in the building of the ARMON installed at the GIC3 station; iii) Albert Jornet, software engineer of the IC3 who developed, together with Roger Curcoll, an automatic system for daily running FLEXPART backward simulations; iv) Israel Lopéz-Coto and the University of Huelva for the

radon flux modelled data; v) We thank Dr. Stefano Galmarini for helping to improve the manuscript. Authors also thank David Carslaw and Karl Ropkins, developers of the R package OpenAir (www.openair-project.org), used in the present work for data analysis.

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
