# Peer review of "Study of the main processes driving atmospheric CH4 variability in a rural Spanish region by Grossi et al. 2017 ACPD."

_Atmospheric Chemistry and Physics, 2017_

## Referee Comment (RC1) · Anonymous Referee #2 · 2 Nov 2017

The paper presents atmospheric data of CH4 mole fractions and Rn222 concentrations observed at a measurement site in central Spain. Surface-atmosphere exchange fluxes of CH4 are estimated based on the radon tracer method, and compared to values from an emission inventory. The topic fits well in the scope of ACP. In general the paper is well written, and I recommend publication after the following concerns have been addressed.

General Comments:

The authors found a strong disagreement of Rn based CH4 flux estimates with the values in the EDGAR inventory. Potential reasons for this should be discussed in more detail. What is the contribution in the regional EDGAR CH4 emissions from different source sectors, e.g. enteric fermentation? Which sector seems to be the main cause

for the disagreement? Discussing such questions would allow for inventory people to better learn from such observationally based estimates.

Footprint calculation: What was used as the height below which particles are assumed to be influenced by surface fluxes? Ln 210 mentions 300 m, but what was assumed in cases with a nocturnal boundary layer height below 300 m? Particles above the top of the nocturnal boundary layer should not be influenced by surface fluxes. If the method assumes all particles below 300 m to be influenced by surface fluxes, the associated uncertainty in the footprint should be described. Note that usually there is strong wind shear near the top of the nocturnal boundary layer, which worsens a potential error in estimated footprint area. Also it is unclear how exactly the weighting function $w(x,t)$ (Eq. 2) was normalized, and what the exact time limits in the summation in Eq. 2 are. This needs to be clearly described.

Please use an equation to better illustrate the FLEXPART Radon-tracer method derived CH4 fluxes (FR_CH4).

Rather than showing a somewhat hard to read map in Fig 1, why not show the footprint map and a map of the inventory based emissions? That would be better related to the rest of the manuscript.

Specific comments

Ln 90: "flux in this area is of about" I suggest to drop the "of"

Ln 124: "The instrument accuracy for CH4 is of 0.36 ppb" I suggest to drop the "of"

Ln 143: Is the canopy really below 20 cm? May be this should read "below 20 m"?

Ln 157: Please rephrase the section header, and avoid unreadable terms (i.e. avoid underline characters).

Ln 177: For which time intervals was the correlation between CH4 and Rn assessed, for a single night? This should be stated

Ln 231: replace "is" by "of"

Ln 242: drop "of"

Ln 243: "it is of 30 ppb" drop the "of"

Fig. 3 and Fig. 4: it would be useful to show the monthly boxplots also separately for day and night, especially for attributing changes in daily amplitudes; it could well be that low nocturnal PBLH drives the larger amplitude during summer rather than the deeper mixing during daytime as stated in Ln 293.

Figure 7: the legend is unnecessary, I suggest removing

Fig. 8: Why are not the monthly values of the UHU climatology shown? Also, it should be mentioned what "local flux" means; is it the UHU Rn flux value of the local pixel containing the GIC3 station?

Ln 336: "is of" drop the "of"

Ln 336: Looking at the red circles in Fig. 9 it seems that the mean should be much lower, somewhere around 0.1 mg CH4 m-2 h-1.

Fig. 9: the grey shaded rectangles seem to be at the wrong position. In the figure caption, e.g. week 21-27 June 2014 is mentioned, while the rectangle seems to be at around mid-end of March 2014. Also, the green shaded rectangle (presence of animals) is located at times with low FR_CH4.

Fig. 10: Please use simple numbers as x-axis labels to indicate the months.

Ln 395-397: this is a repetition of Ln 287-289

Ln 404-405: I disagree with the assumption that CH4 fluxes vary only to a small degree; this has not been shown. In Ln 390 the authors even argue that the hysteresis in Fig. 5 is due to changes in local emissions. I suggest citing literature describing the emissions from animals; what is expected from the process level, e.g. do ruminants

emit constantly, or more during certain parts of their diurnal feeding cycle?

---

## Referee Comment (RC2) · Anonymous Referee #1 · 2 Feb 2018

General comments

The article reports three years of atmospheric methane (CH4) and radon (222Rn) measurements at a measurement site in central Spain, and presents an analysis of the variability of these two parameters for the period of three years (2013-2015), including also modelled 222Rn fluxes and planetary boundary layer heights. Furthermore, nocturnal CH4 fluxes were estimated using the radon tracer method (RTM) as well as a bottom-up emission inventory.

The text is written clearly enough, but should be further improved - best revised by a native speaker/writer (e.g. to improve the structure of sentences). Figures 1 to 3 are too small and the legends as well as labels of Figs. 1 to 2 are not legible. Figure 2S is much too crowded with labels and not well legible. I am not convinced by the color

scale used in figures 5 to 7; is this safe for color-blind readers? Particularly in Fig. 5, the colors for hours 5 to 8 look practically the same.

I agree with the comment by Referee #2 regarding the disagreement of 222Rn-based CH4 flux estimates with the EDGAR inventory-based ones. While it might well be that livestock is responsible at least for a part of the CH4 signal, I failed to see a proof in this work. Moreover, EDGAR should be sensitive to livestock emissions (as they are non-natural), but the opposite seems to be the case. This seems to indicate that the main processes driving CH4 variability at GIC3 area are natural ones or that EDGAR is performing poorly at least when livestock is concerned. In my opinion, the focus, discussion and conclusions of the article should be more on the method and less trying to link the CH4 variability mostly to livestock as it is the case in the current version. In this context, I also find the title of the article a bit ill chosen.

The section 2.2 is very minimalistic. I acknowledge that concise descriptions of measurement systems is not in the scope of articles in ACP, but as there is no other reference to direct the reader to, at least a schematic of the measurement setup could be added in the Supplement. In my opinion, the paper is suitable for publication in ACP, but only if the comments have been addressed properly.

Specific comments and technical corrections

Note on Technical corrections: in some cases, I have marked a word or formatting only once, but make sure to apply the corrections throughout the text where relevant.

Line 17 (L 17): instead of "concentration" use rather "(dry air) mixing ratio". Sentence is too long and difficult to read/understand.

L 21: delete "previous"

L 27: delete "of" in "is of 0.32"

L 36: reported by whom?

L 49: "….data and data products…"

L 51: "In some European regions…."

L 52: what do you mean by "remote"? Please define this more clearly.

L 64: "In this study, we analyzed the time series…...and December 2015."

L 68: delete "Particularly,"

L 69: delete "such as Extremadura" – you mention it in L 72 again.

L 75: delete "further"; better replace "mobile" with "ephemeral" or "transient" (without the quotes in the text)

L 83-85: are the durations of the cold and warm seasons defined anywhere in the text?

L 91: "The GNP is located in a granitic basement;"? Rather: "The GNP has a (predominantly) granitic basement and is thus covered by granitic soils with high …."

Fig. 1: missing unit in the legend, add reference for CORINE/the map (…., 2007)

L 98: delete "Particularly,"

L 100: "In Figure 2, a map …"

Fig. 2: instead of "Source", use "Modified from"

L 120: the reference "Crosson, 2008" is not well chosen here – it would be better to leave it out. Change to "… measured with a frequency ….using a…"

L 125: a target gas is, more precisely, used for "checking the stability and quality of the instrument calibration". Please define better what you mean by "according to the definitions of the World Meteorological Organization (WMO)."; add a reference.

L 131: "…of both ARMON and G2301 analyzer are…"

L 134: Sample air drying system

L 144: "...area is quite hilly." is not very explicit, please elaborate on this. A figure showing the terrain would be helpful for understanding to what extent it is justified to apply a method as RTM at GIC3 (c.f. assumptions in Lines 160 to 175).

L 149: how representative are the ECMWF PBLH data for the GIC3 site? This question also relates to previous comment (L 144) – is the variability of the terrain captured well enough in the ECMWF model?

L 185: please explain the acronym UHU

L 195: "...country on a spatial grid."

L 196: provides global annual CH4 emissions on a 0.1 degree resolution

L 225: "...sample system 11 % of the..." How are the data gaps distributed; evenly or was there a concentration of data gaps in some periods /in which ones?

Fig. 3 I presume "Hour of the day" is in UTC? Please add. Also, better use nmol/mol instead of concentration, which should only be used in communicating with the general public (see e.g. GAW Report No. 229)

L 245: I cannot follow this sentence "A light increase of methane concentrations seems to be observed between the first and the second semester of the year." – please clarify

L 305: delete "Indeed,"

Fig. 9: correct the month name abbreviations to English language; green circles are poorly visible

L 392: if CH4-enhanced air masses were transported in the afternoon, would we not see the same pattern for Rn as well? Please elaborate on this in more detail. It would be interesting to actually see a typical footprint for such events.

L410: There was not much said on the landscape, precipitation patterns, water (bodies), etc. in the region – it is a reasonable guess that livestock has something to do

with it, but there might be other reasons for this increase in CH4 fluxes - this should be discussed.

---

## Author Comment (AC3) · 26 Mar 2018

We thank the two anonymous reviewers for their comments on our manuscript. Please find a point-by-point reply to each of the comments raised by the referees

Regards Claudia Gross
* * *

---

## Author Response (AR1)

**Dear Editor,**

First of all, we would like to thank the reviewers for their comments and suggestions which have been really useful to improve our work.

The corrections suggested by both reviewers, listed in chronological order, have been applied as here reported (you will find our answers in blue):

**Anonymous Referee #2 (Received and published: 2 November 2017)**

General Comments: The authors found a strong disagreement of Rn based  $CH_4$  flux estimates with the values in the EDGAR inventory. Potential reasons for this should be discussed in more detail. What is the contribution in the regional EDGAR  $CH_4$  emissions from different source sectors, e.g. enteric fermentation? Which sector seems to be the main cause for the disagreement? Discussing such questions would allow for inventory people to better learn from such observationally based estimates.

Thank you for highlighting this point. Given that the EDGAR  $CH_4$  emissions are provided on an annual scale, we would like to underline the fact that the main aim of this work is to show how Rn-based  $CH_4$  flux estimates can offer information on 'seasonal sources'. These can be anthropogenic sources too, but with seasonal behaviour (e.g. agricultural activity), which are not captured in EDGAR or classical UNFCCC inventories. Although we observed that annual mean Rn-based  $CH_4$  flux estimates are lower than the values based on the EDGAR inventory over the study period, we were much more interested in understanding possible reasons for the relative increase and/or decrease of these differences during two semesters of the year (June-December and January-May) (Figures 9 and 11 of the manuscript).

In the results paragraph of our revised manuscript we have now commented on the possible reasons for the observed disagreement between the two methods and we have also carried out a second experiment using a comparison factor, coming from another 222Rn emission product, to rescale our results. We find that the disagreement with EDGAR is mainly reduced, while the seasonal amplitude of the RTM-based CH4 emissions is enhanced. The differences between Rn-based CH4 flux estimates and values based on the EDGAR inventory could be mainly due to:

1) applied RTM methodology:

A possible underestimation of the 222Rn flux data used within the RTM. The outputs from the UHU radon flux model will lead to lower FR\_CH4 fluxes if they are lower than actual 222Rn fluxes (Equation 2). Karstens et al., 2015 compared their radon flux model with UHU model and they found a generally 40 % higher222Rn exhalation rate in their map than in the López-Coto et al. (2013) map. The 40% factor observed by Karstens et al., 2015 has been applied in our study to calculate rescaled FR\_CH4 values (FR\_CH4-rescale). In Figure 11 boxplot of the modified manuscript monthly medians of these values have been compared with FE\_CH4 and FR\_CH4 fluxes. FR\_CH4-rescale fluxes show a good agreement with FE\_CH4 fluxes during the months between June and December, when the transhumant livestock stays in the GIC3 area. A further validation of both 222Rn flux models should be carried out with high spatial resolution over Europe as suggested by Karstens et al., 2015.

**2) Spatial and temporal disaggregation in EDGAR:**

The mean contribution in the regional EDGAR  $CH_4$  emission of the enteric fermentation is 38% of the total (EDGARv4.2, 2010). The spatial distribution of these emissions over the country in the EDGARV4.2 methodology (http://edgar.jrc.ec.europa.eu/methodology.php) was built up

using spatial proxy datasets with the location of energy and manufacturing facilities, road networks, shipping routes, human and animal population density and agricultural land use, which vary over time. National sector totals are then distributed with the given percentages of the spatial proxies over the country's area. This could lead to the assignment of higher emissions in some regions such as the GIC3 area if transhumant cattle are fully taken into account.

The fact that the RTM and EDGAR results are in better agreement during the month when cattle are present could suggest that the inventory did attribute emissions there when scaling annual totals. Actually, The Unión de Pequeños Agricultores (UPA, 2009) reports that between 2004 and 2009 an average of 800,000 transhumant animals were hosted in Spain and 40,000 (5% of total) were counted in the province of Ávila (extension: 8050.15 km2) for an average of 5 cows per square km where the GIC3 station is located and their whereabouts can be expected to change local/regional CH4 emissions when they are present/moving in a region.

**3) Systematic/seasonal bias in footprint calculations**

To estimate the impact of the EDGAR emissions for the GIC3 region, we rely on footprints calculated using ECMWF-FLEXPART. If the surface sensitivity calculated in the model is systematically biased (lower) compared to the real sensitivity, the FE\_CH4 fluxes could be underestimated. Even slight seasonal changes of model performance could be possible due to the fixed PBLH scheme (300m). If the true PBLH was below 300m during winter we would overestimate the impact of emissions as particles above the PBLH, but below 300m would still be assumed to be impacted by emissions. Another point to consider is that the night-time PBLH does not show strong seasonal change (see Figure 4b). The sudden increase in CH4 emissions during the period when transhumant cattle reach the GIC3 regions cannot be explained by this, as the models ability to represent atmospheric conditions should not change from one week to another, given that general meteorological conditions do not change on this time-scale, see radon and met data in Grossi et al 2016. Finally, RTM and EDGAR methodologies are based on the same footprints so this effect should not influence the relative differences observed by Cattle and No-Cattle seasons.

Footprint calculation: What was used as the height below which particles are assumed to be influenced by surface fluxes? Ln 210 mentions 300 m, but what was assumed in cases with a nocturnal boundary layer height below 300 m? Particles above the top of the nocturnal boundary layer should not be influenced by surface fluxes. If the method assumes all particles below 300 m to be influenced by surface fluxes, the associated uncertainty in the footprint should be described. Note that usually there is strong wind shear near the top of the nocturnal boundary layer, which worsens a potential error in estimated footprint area.

We made the common assumption in FLEXPART of a fixed height layer to calculate the footprint or source-receptor relationship (e.g. Stohl et al. 1998, Pan et al. 2014). A PBLH cut-off of 300m was assumed for the calculation of the footprints using 24h back-trajectories and waiting for the particles to pass over the footprint (Equation 3 and 4 of the revised manuscript). Although this selection could introduce an error in the estimation of the residence time within the nocturnal boundary layer, this residence time is used to calculate both FE\_\_CH4 and the effective 222Rn flux (used to calculate the FR\_CH4 fluxes, see equation 2 of the revised manuscript). In addition, night-time PBLH at GIC3 does not show strong seasonality (see Figure 4a in manuscript).

We have added this information in the methodology section and discussed its influence on the results in the discussion.

Also it is unclear how exactly the weighting function w(x,t) (Eq. 2) was normalized, and what the exact time limits in the summation in Eq. 2 are. This needs to be clearly described.

We have added this, as suggested (Equation 4).

Please use an equation to better illustrate the FLEXPART Radon-tracer method derived CH4 fluxes (FR\_CH4).

It has been added as suggested (Equation 2).

Rather than showing a somewhat hard to read map in Fig 1, why not show the footprint map and a map of the inventory based emissions? That would be better related to the rest of the manuscript.

Thanks for this suggestion. We have now added the footprint and EDGAR inventory maps (new Figures 1 and 2) within the manuscript and the map of the transhumance paths was moved to the supplement material (new Figure S2).

Specific comments

Ln 90: "flux in this area is of about" I suggest to drop the "of"

This has been changed.

Ln 124: "The instrument accuracy for CH4 is of 0.36 ppb" I suggest to drop the "of"

This has been changed

Ln 143: Is the canopy really below 20 cm? May be this should read "below 20 m"?

Yes, it was 20m - thanks. This has been changed

Ln 157: Please rephrase the section header, and avoid unreadable terms (i.e. avoid underline characters).

**This has been changed**

Ln 177: For which time intervals was the correlation between CH4 and Rn assessed, for a single night? This should be stated.

This was stated in Section 2.4.1 when the radon tracer methodology was presented. We have changed this sentence to clarify it.

Ln 231: replace "is" by "of"

This has been changed

Ln 242: drop "of"

This has been changed

Ln 243: "it is of 30 ppb" drop the "of"

It has been changed

Fig. 3 and Fig. 4: it would be useful to show the monthly boxplots also separately for day and night, especially for attributing changes in daily amplitudes; it could well be that low nocturnal PBLH drives the larger amplitude during summer rather than the deeper mixing during daytime as stated in Ln 293.

The additional results have been added and discussed in the results paragraph.

Figure 7: the legend is unnecessary, I suggest removing

This has been changed

Fig. 8: Why are not the monthly values of the UHU climatology shown? Also, it should be mentioned what "local flux" means; is it the UHU Rn flux value of the local pixel containing the GIC3 station?

Monthly UHU values are not shown in this plot because they were already shown in Figure 7. The local flux is actually the UHU Rn flux value of the local pixel containing the GIC3 station. This has been pointed out within the manuscript.

Ln 336: "is of" drop the "of"

This has been changed

Ln 336: Looking at the red circles in Fig. 9 it seems that the mean should be much lower, somewhere around 0.1 mg CH4 m-2 h-1.

The reviewer is right, there was an editing error. The value was 0.13 and this has been corrected.

Fig. 9: the grey shaded rectangles seem to be at the wrong position. In the figure caption, e.g. week 21-27 June 2014 is mentioned, while the rectangle seems to be at around mid-end of March 2014. Also, the green shaded rectangle (presence of animals) is located at times with low FR\_CH4.

The reviewer is right, there was an error in the plot because the shaded boxes moved. This has been corrected.

Fig. 10: Please use simple numbers as x-axis labels to indicate the months.

This has been changed.

Ln 395-397: this is a repetition of Ln 287-289

We have deleted the repeated sentence.

Ln404-405: I disagree with the assumption that CH4 fluxes vary only to a small degree; this has not been shown. In Ln 390 the authors even argue that the hysteresis in Fig. 5 is due to changes in local emissions. I suggest citing literature describing the emissions from animals; what is expected from the process level, e.g. do ruminants emit constantly, or more during certain parts of their diurnal feeding cycle?

We have extended and improved this section in the discussion. We agree with the reviewer that the CH4 fluxes can also vary on a diurnal cycle. The hysteresis observed in Figure 5 which could be due to changes in local emissions appears between 13.00-18.00 UTC, which cannot be tracked using the RTM. Although some studies have found strong diurnal changes in ruminant emissions, e.g. Bilek et al. 2001, Wang et al., 2015, these studies link the diurnal pattern of methane emissions to the ruminant feeding cycle in feedlots. They find that the feeding regime, feeding frequency and the amount of feed offered can alter methane emissions. Given that transhumant cattle are moved to the GIC3 region to graze, we would not assume that this effect is as pronounced as in feedlots, as cattle can feed more continuously at GIC3. Mohammed, et al. (2011) reported a fairly flat daily cycle of  $CH_4$  emissions from grazing, especially if compared to aforementioned feedlot studies. However, we actually do not have any direct information about the feeding cycle of grazing Gredos livestock, but we now mention this as a future step in the identification of methane emission in this area in the discussion.

**Anonymous Referee #1 (Received and published: 2 February 2018)**

The text is written clearly enough, but should be further improved - best revised by a native speaker/writer (e.g. to improve the structure of sentences).

Thanks, the text has now been corrected by a native English writer.

Figures 1 to 3 are too small and the legends as well as labels of Figs. 1 to 2 are not legible

Figure 1 has been deleted and Figure 2 has been moved to the supplement, as suggested by referee #2.

Figure 2S is much too crowded with labels and not well legible.

Figure 2S, now 4S, has been changed as suggested.

I am not convinced by the color scale used in figures 5 to 7; is this safe for color-blind readers? Particularly in Fig. 5, the colors for hours 5 to 8 look practically the same.

We have tried to make the plot acceptable for all color-blind readers but we finally decided to use the first version of the plot because the whole paper has colored figures. We have avoided green as most colorblindness falls on the green-red spectrum (deuteranopia).

I agree with the comment by Referee #2 regarding the disagreement of 222Rn-based CH4 flux estimates with the EDGAR inventory-based ones. While it might well be that livestock is responsible at least for a part of the CH4 signal, I failed to see a proof in this work.

The possible reasons for this disagreement have been added in the discussion and a detailed explanation has been given above (reply to Referee #2).

Moreover, EDGAR should be sensitive to livestock emissions (as they are non-natural), but the opposite seems to be the case. This seems to indicate that the main processes driving CH4 variability at GIC3 area are natural ones or that EDGAR is performing poorly at least when livestock is concerned. In my opinion, the focus, discussion and conclusions of the article should be more on the method and less trying to link the CH4 variability mostly to livestock as it is the case in the current version. In this context, I also find the title of the article a bit ill chosen.

Our interpretation of the findings is not that EDGAR performs poorly for the livestock component, as the different methods are in reasonable agreement during the period when livestock is present in the GIC3 region mainly using the RTM\_CH4\_rescale. Our results seem to show that the RTM-based CH4 fluxes decrease during the period without transhumant livestock in the GIC3 area and they increase during June-December when the livestock is back to the region. On the contrary, the EDGAR based CH4 fluxes do not show any seasonality. Thus, it seems more likely that all (annual) CH4 emissions of these cattle have been attributed to this region, although they are physically moved to different regions. Given the scope of EDGAR we would not expect it to cover all local processes and this study intends to help identify transhumance as a potential issue that could be improved (added) in future emission inventories for this region and Spain as a whole. However, we agree with the reviewer that the conclusions of our work should be more focused on the applied method and the paragraph has been changed accordingly. The title of the article has been changed to better fit with the work done

The section 2.2 is very minimalistic. I acknowledge that concise descriptions of measurement systems is not in the scope of articles in ACP, but as there is no other reference to direct the reader to, at least a schematic of the measurement setup could be added in the Supplement.

In agreement with the reviewer's suggestion, a schematic diagram of the measurement set-up used at the GIC3 station has been added as Figure S3 in the supplement.

Specific comments and technical corrections

Note on Technical corrections: in some cases, I have marked a word or formatting only once, but make sure to apply the corrections throughout the text where relevant.

Line 17 (L 17): instead of "concentration" use rather "(dry air) mixing ratio". Sentence is too long and difficult to read/understand.

The sentence has been changed as, suggested by the reviewer.

L 21: delete "previous" done

- L 27: delete "of" in "is of 0.32" done
- L 36: reported by whom? 'by each country' has been added

L 49: "....data and data products..." done

L 51: "In some European regions...." done

L 52: what do you mean by "remote"? Please define this more clearly. This has been changed to 'with stations located in natural parks'

L 64: "In this study, we analyzed the time series.....and December 2015." this has been changed

L 68: delete "Particularly," done

L 69: delete "such as Extremadura" - you mention it in L 72 again. done

L 75: delete "further"; better replace "mobile" with "ephemeral" or "transient" (without the quotes in the text) done

L 83-85: are the durations of the cold and warm seasons defined anywhere in the text? This has been done now.

L 91: "The GNP is located in a granitic basement;"? Rather: "The GNP has a (predominantly) granitic basement and is thus covered by granitic soils with high ...." Fig. 1: missing unit in the legend, add reference for CORINE/the map (...., 2007)

This has been changed

L 98: delete "Particularly," done

L 100: "In Figure 2, a map ..." done

Fig. 2: instead of "Source", use "Modified from" done

L 120: the reference "Crosson, 2008" is not well chosen here – it would be better to leave it out. Change to "... measured with a frequency ....using a..."

done

L 125: a target gas is, more precisely, used for "checking the stability and quality of the instrument calibration". Please define better what you mean by "according to the definitions of the World Meteorological Organization (WMO)."; add a reference.

A definition and appropriate reference have been added

L 131: "...of both ARMON and G2301 analyzer are..." done

L 134: Sample air drying system done

L 144: "...area is quite hilly." is not very explicit, please elaborate on this. A figure showing the terrain would be helpful for understanding to what extent it is justified to apply a method as RTM at GIC3 (c.f. assumptions in Lines 160 to 175).

A figure showing the GIC3 topography has been added as Figure S1 in the supplement.

L149: how representative are the ECMWF PBLH data for the GIC3 site? This question also relates to previous comment (L 144) – is the variability of the terrain captured well enough in the ECMWF model?

Seidel et al., 2012 found that compared with radiosonde observations, both the re-analysis and the climate models produce deeper layers due to the difficulty in simulating stable conditions. In vertical profiles they introduce height uncertainties that can exceed 50% for shallow boundary layers (<1 km), but are generally <20% for deeper boundary layers. This information has been added to the revised manuscript.

L 185: please explain the acronym UHU - done

L 195: "...country on a spatial grid."- done

L 196: provides global annual CH4 emissions on a 0.1 degree resolution - done

L 225: "...sample system 11 % of the..." How are the data gaps distributed; evenly or was there a concentration of data gaps in some periods /in which ones? This information has been added. We mainly missed summer 2013.

Fig. 3 I presume "Hour of the day" is in UTC? Please add. Also, better use nmol/mol instead of concentration, which should only be used in communicating with the general public (see e.g. GAW Report No. 229). - done

L245: I cannot follow this sentence "A light increase of methane concentrations seems to be observed between the first and the second semester of the year." – please clarify. This has been clarified in the manuscript

L 305: delete "Indeed," - done

Fig. 9: correct the month name abbreviations to English language; green circles are poorly visible - done

L 392: if CH4-enhanced air masses were transported in the afternoon, would we not see the same pattern for Rn as well? Please elaborate on this in more detail. It would be interesting to actually see a typical footprint for such events.

If air masses rich in methane, but not in radon, are transported to the station, we will not be able to see the same daily pattern in radon concentration. We have tried to explain this effect now within the manuscript using Figures 9, 10 and S4 of the supplement, where an increase of the methane fluxes when air masses are coming from the Madrid direction is shown. L410: There was not much said on the landscape, precipitation patterns, water (bodies), etc. in the region - it is a reasonable guess that livestock has something to do with it, but there might be other reasons for this increase in CH4 fluxes - this should be discussed

We have added this in the conclusions paragraph.

|    | **                                                                                                                                                                                                                                                                                                                                                                                                  |       | Style Definition                           |         |
|----|-----------------------------------------------------------------------------------------------------------------------------------------------------------------------------------------------------------------------------------------------------------------------------------------------------------------------------------------------------------------------------------------------------|--------------|--------------------------------------------|---------|
|    | Study of the main processes drivingdaily and seasonal                                                                                                                                                                                                                                                                                                                                               | $\backslash$ | Formatted: Left: 3 cm, Right: 3            |
cm. |
|    | atmospheric CH 4 mixing ratio variability in a rural Spanish                                                                                                                                                                                                                                                                                                                      |              | Top: 2.5 cm, Bottom: 2.5 cm, He
29.7 cm | ight:   |
|    | region using 222 Rn tracer                                                                                                                                                                                                                                                                                                                                                               |              | Formatted: Font: 10 pt, English (          | (U.S.)  |
| 5  | C. Claudia Grossi a.1. F.2 , Felix R. Vogel b , R. Roger Curcoll a.2 , I. López Cotoe.3 , A. Alba Àgueda a.4 , A.
Vargas d , X. Arturo Vargasc, Xavier Rodó a.ed.3.5 , J. A. Josep-Anton Morguí a.f.2e.3 | /            | Formatted                                  |         |
|    | a Institut Català de Ciències del Clima (IC3), Barcelona, Spain                                                                                                                                                                                                                                                                                                                   |              | Formatted                                  |         |
| 10 | Paris Saclay, Gif sur Yvette, France.                                                                                                                                                                                                                                                                                                                                                               |              |                                            |         |
|    | e <del>Departamento de Física. Universidad de Huelva (UHU). Spain</del>                                                                                                                                                                                                                                                                                                                  |              |                                            |         |
|    | 4b Climate Research Division. Environment and Climate Change Canada, Toronto, Canada                                                                                                                                                                                                                                                                                                     |              |                                            |         |
|    | L'Institut de Tècniques Energètiques (INTE), Universitat Politècnica de Catalunya (UPC), Barcelona,                                                                                                                                                                                                                                                                                                 |              | Formatted                                  |         |
|    | Spain                                                                                                                                                                                                                                                                                                                                                                                               | $\square$    |                                            |         |
| 15 | e Instituciò d Institució Catalana de Recerca i Estudis Avançats (ICREA), Barcelona, Spain                                                                                                                                                                                                                                                                                    |              | Formatted: Font: Italic, Spanish           |         |
|    | € Departament <del>d'Ecologia</del> Biologia Evolutiva, Ecologia i Ciències Ambientals, Universitat de Barcelona                                                                                                                                                                                                                                                                         |              | (International Sort), Superscript          |         |
|    | (UB), Barcelona, Spain                                                                                                                                                                                                                                                                                                                                                                              |              | Formatted: Font: Italic, Spanish           |         |
|    | Present addresses:                                                                                                                                                                                                                                                                                                                                                                                  |              | (International Sort)                       |         |
|    | 1 Institut de Tècniques Energètiques (INTE), Universitat Politècnica de Catalunya (UPC), Barcelona,                                                                                                                                                                                                                                                                                      |              | (International Sort), Superscript          |         |
| 20 | Spain                                                                                                                                                                                                                                                                                                                                                                                               |              | Formatted: Font: Italic, French            |         |
|    | 2 Institut 2 Physics Department, Universitat Politècnica de Catalunya (UPC), Barcelona, Spain                                                                                                                                                                                                                                                                                 |              | (Canada), Superscript                      |         |
|    | 3 Institut de Ciència i Tecnologia Ambientals (ICTA), Universitat Autònoma de Barcelona (UAB),                                                                                                                                                                                                                                                                                           | 1            | Formatted                                  |         |
|    | Cerdanyola del Vallès, Spain                                                                                                                                                                                                                                                                                                                                                                        | /            |                                            |         |
|    | 3 National Institute of Standards and Technology (NIST), Gaithersburg, MD, US                                                                                                                                                                                                                                                                                                            |              |                                            |         |
| 25 | 4 Centre d'Estudis del Risc Tecnològic, Universitat Politècnica de Catalunya (UPC) -                                                                                                                                                                                                                                                                                              | 1            | Formatted                                  |         |
|    | BarcelonaTech, Barcelona, Spain                                                                                                                                                                                                                                                                                                                                                                     |              |                                            |         |
|    | 5 CLIMA2, Climate and Health Program, ISGlobal (Barcelona Institute of Global Health), Barcelona,                                                                                                                                                                                                                                                                                        | 1            | Formatted                                  |         |
|    | Spain                                                                                                                                                                                                                                                                                                                                                                                               |              |                                            |         |
|    | Correspondence to: Claudia Grossi (claudia.grossi@upc.edu), Felix Vogel                                                                                                                                                                                                                                                                                                                             | 1            | Formatted                                  |         |
| 30 | (felix.vogel@ <del>lsce.ipsl.frcanada.ca)</del>                                                                                                                                                                                                                                                                                                                                              |              |                                            |         |
|    | Abstract. Atmospheric concentrations of the two main greenhouse gases (GHGs), The ClimaDat station                                                                                                                                                                                                                                                                                                  |              |                                            |         |
|    | at Gredos (GLC3) has been continuously measuring atmospheric (dry air) mixing ratios of carbon dioxide                                                                                                                                                                                                                                                                                              |              |                                            |         |
|    | $(CO_2)$ and methane $(CH_4)$ , are continuously measured since November 2012 at the Spanish rural station
of Creates $(CIC2)$ , within the elimeter rates of Cline Data transition in the interval $(222)$                                                                                                                                                                                      |              |                                            |         |
| 25 | or Greaos (GIC3), within the climate network ClimaDat, together with atmospheric radon (  Rn) tracer                                                                                                                                                                                                                                                                                     |              |                                            |         |
| 33 | anu as wen as meteorological parameters. Ine., since November 2012. In this study we investigate the                                                                                                                                                                                                                                                                                                |              | Formatted: Footer, Right                   |         |

atmospheric variability of  $CH_4$  concentrations measured frommixing ratios between 2013 toand 2015 at GIC3 has been analyzed in this study. It is interpreted in regard to the variability of measured atmospherie with the help of co-located observations of 222Rn concentrations, modelled 222Rn fluxes and modelled heights of the planetary boundary layer heights (PBLH) for the same period. In addition, nocturnal fluxes of  $CH_4$  were estimated using two methods: the Radon Tracer Method (RTM) and one based on the application of the EDGARv4.2 bottom up emission inventory.). Both previous methods have

en applied using the same footprints, calculated by the atmospheric transport model FLEXPARTv6.2.

Results show that daily and seasonal changes in atmospheric CH4 can be better understood with the help 45 of atmospheric concentrations of 222Rn (and its the corresponding fluxes) can help to understand the atmospheric CH4 variability.). On a daily basistimescale, the variation in the PBLH mainly drives changes inis the main driver for 222Rn and CH4 concentrations variability while, on monthly basistimescales, their atmospheric variability seems to be due todepend on emission changes in their. To understand (changing) CH4 emissions-Median, nocturnal fluxes of CH4 were estimated using two methods: the Radon Tracer 50 Method (RTM) and a method based on the EDGARv4.2 bottom-up emission inventory using FLEXPARTv9.0.2 footprints. The mean value of RTM—-based methane fluxes (FLEXPART\_RTMFR\_CH4) is 0.1011 mg CH4 m-2 h-1 with a standard deviation of 0.09 mg CH4 m-2 h-1. Median- or 0.29 mg CH4 m-2 h-1 with a standard deviation of 0.23 mg CH4 m-2 h-1 when using a rescaled 222Rn map (FR\_CH4\_rescale). For our observational period, the mean value of methane fluxes based on

- 55 the bottom-up inventory (FLEXPART EDGARFE\_CH4) is of 0.3233 mg CH4 m-2 h-1 with a standard deviation of 0.08 mg CH4 m-2 h-1. The FLEXPART\_EDGAR\_Monthly CH4 fluxes due to the contribution of the cities in the GIC3 region present a median value of 0 mg CH4-m-2 h-4 with a standard deviation of 0.06 mg CH4-m-2 h-4. Monthly FLEXPART\_RTM\_CH4-flux shows based on RTM (both FR CH4 and FR CH4 rescale) show a seasonality which is not observed in thefor monthly FLEXPART\_EDGAR\_CH4
- flux. Actually, a minimum duringFE\_CH4 fluxes. During January-May and a maximum, RTM-based CH4
   fluxes present mean values 25% lower than during June-December-are observed in these first fluxes...
   This previous variability seems to be mainly related to the alternate presence seasonal increase of methane
   fluxes calculated by RTM for the GIC3 area appears to coincide with the arrival of transhumant livestock
   in the GIC3 area. The results obtained in this study should be further investigated using longer CH4 and
   222Rn time series to obtain more robust statistics and help to improve the seasonality of the emission
  - factors from bottom up inventories at GIC3 in the second semester of the year.

Keywords: methane, flux, radon, atmosphere, livestock, EDGAR, FLEXPART.

**Introduction**

40

70 The importanceimpact of the atmospheric increase of the greenhouse gases (GHGs) foron climate change processes is well known (IPCC, 2013). Therefore, GHGs emissions, due to natural as well as anthropogenic sources, are currently estimated and reported by each national agency to the United Nations Framework Convention on Climate Change (UNFCCUNFCCC). A goodbetter understanding of

|     | the underlying processes causing thethese emissions can help in the implementation of tuture emission                          |   |                                        |   |
|-----|--------------------------------------------------------------------------------------------------------------------------------|---|----------------------------------------|---|
| 75  | reduction strategies. Among the GHGs covered under the UNFCCC tramework methane (CH 4 )                             |   |                                        |   |
|     | is the second most important anthropogenic GHG- that is covered by the UNFCCC. The atmospheric                                 |   |                                        |   |
|     | concentration mixing ratio of $CH_4$ has substantially changed since pre-industrial times from a global                 |   |                                        |   |
|     | average of 715 ppbnmol mol -1 to more than 1774 ppbnmol mol -1 (IPCC, 2013). TodayNowadays, the          |   | Formatted: Superscript                 | J |
|     | contribution of $CH_4$ related to anthropogenic activities in the atmosphere represents about 25% of the                       |   |                                        |   |
| 80  | total additional anthropogenic radiative forcing (IPCC, 2013). However, CH 4 has a relatively short                 |   |                                        |   |
|     | lifetime in the atmosphere (~ 9 years) and this makes it relevant forin defining immediate and efficient                       |   |                                        |   |
|     | emission reduction measuresstrategies (Prinn et al., 2000). Particularly, in Spain, man-made methane                           |   |                                        |   |
|     | emissions are mainly due to enteric fermentation (3138%), management of manure (20%), and landfills                            |   |                                        |   |
|     | (36%) (WWF, 2014; MMA, 2016). The remaining methane contributions in Spain are due to rice                                     |   |                                        |   |
| 85  | cultivation (e.g. Àgueda et al., 2017), coal mining, leaks in natural gas infrastructureinfrastructures and                    |   |                                        |   |
|     | waste water treatment-related processes. The CH4 emission due to enteric fermentation related to                               |   |                                        |   |
|     | livestock is directly linked to the number of animals of each type/breed of cattle, their age, their diet and                  | I |                                        |   |
|     | environmental conditions (MMA, 2016). Spanish CH 4 emissions for 2014 due to enteric fermentation                   | 1 |                                        |   |
|     | were estimated to be $\frac{11,704}{11,704}$ Gg CO 2 -eq (MMA, 2016).                                    |   | Formatted: Font: 10 pt, English (U.S.) | ) |
| 90  | *                                                                                                                              |   | Formatted: English (U.S.)              | ĺ |
|     | In order to estimate CHCs emissions, bettern up (based on fuel consumption and enthronogenic estivity)                         |   | Formatted: Normal (Web), Justified,    | ĺ |
|     | deta) and tan down methods (head on streagheric shormations and modelling) are both widely and id                              |   | Line spacing: 1.5 lines                | J |
|     | data) and top-down methods (based on atmospheric observations and modeling) are both widely applied                            | 1 |                                        |   |
|     | and the scientific community is focusinghas focussed on reducing their related uncertainties and                               |   |                                        |   |
|     | understanding systematic inconsistencies (e.g. Vermeulen et al., 2006; Bergamaschi et al., 2010; NRC,                          |   |                                        |   |
| 95  | 2010; Jeong et al., 2013; Hiller et al., 2014). Top-down methods usually require both high-quality and                  |   |                                        |   |
|     | long-term GHGs observations. European projects, such as InGOS (www.ingos-infrastructure.eu), and                               |   |                                        |   |
|     | infrastructures, such as ICOS (www.icos-infrastructure.eu), aim to offer atmospheric CO 2 and non-                  |   | Formatted: Internet Link, Font: 12 pt  | J |
|     | CO2 GHGs data and data products to better understand GHGGHGs fluxes in Europe and adjacent regions.                            |   | Formatted: Font: 10 pt, English (U.K.) | ) |
|     |                                                                                                                                | 1 |                                        |   |
| 100 | NeverthelessUnfortunately, in southernsome European regions, such as Spain, there is still a significant                       | 1 |                                        |   |
|     | lack of high-quality atmospheric GHGs observations. The Catalan Institute of Climate Sciences (IC3) has                        | I |                                        |   |
|     | been working since 2010 within the ClimaDat project at the creation of in setting up a network of remote                       | 1 |                                        |   |
|     | stations in national parks for continuous measurements of mixing ratios of GHGs, tracers and                                   |   |                                        |   |
|     | meteorological parameters (www.climadat.es). The IC3 network mainly aims to monitor and study the                              | I |                                        |   |
| 105 | exchange of GHGs between the land surface and the lower atmosphere (troposphere) in different                                  |   |                                        |   |
|     | ecosystems, which are characterized by different biogenic and anthropogenic processes, under different                         |   |                                        |   |
|     | synoptic conditions.                                                                                                           |   | Formatted: Font: 10 pt English (ILK)   | h |
|     |                                                                                                                                | Γ | romated rom to pt, English (c.i.t.)    | ļ |
|     | Pasidas CHCs concentrationemining ratios, as leasted observations of additional gases can provide us                           |   |                                        |   |
| 110 | with useful tracers for source experiment studies on to help us to hetter understand etmospheric                               |   |                                        |   |
| 110 | with user in factors for source apportionment studies of to help us to better understand atmospheric $\frac{222}{2}$    |   |                                        |   |
|     | processes (e.g. Zanorowski et al., 2004). Farticularly the The radioactive noble gas radon ( 226 Rn), due to |   |                                        |   |
|     | its chemical and physical characteristic (e.g. Nazaroff and Nero, 1988), is being extensively used for                         |   |                                        |   |
|     | studying atmosphere dynamics, such as boundary layer evolution, (e.g. Galmarini, 2006, Vinuesa and                             |   | Formatted: Footer, Right               | J |
|     | 3+                                                                                                                             | 1 |                                        |   |

Galmarini, 2007), and soil-atmosphere exchanges (e.g. Schery et al., 1998; Zahorowski et al., 2004;115Szegvary et al., 2009; Grossi et al., 2012; Vargas et al., 2015; Grossi et al., 2016). European GHGsmonitoring infrastructures are-already includingincludeatmospheric 222Rn monitors in their stations (e.g.
Arnold et al., 2010; Zimnoch et al., 2014; Schmithüsen et al., 2016). The co-evolution of atmospheric222Rn and GHGs concentrations can also be used inwithin
local/regional GHGs fluxes (e.g. Van der Laan et al., 2010; Levin et al. 2011; Vogel et al. 2012; Wada et

120 al., 2013; Grossi et al., 2014).

In this study we analysed the new-time series of atmospheric CH4 concentrations mixing ratios measured at the IC3 station of Gredos and Iruelas (GIC3) between January 2013 and December 2015-has been analyzed. The main aim was to investigate the major causes influencingmain drivers that influence the 125 daily and seasonal variability of methane concentrations in ethis mountainous rural southern European region. The GIC3 station is located on the Spanish plateau, an area mainly characterized by livestock activity and where the transhumance is still practiced (Ruiz Perez and Valero Sáez, 1990). This is an ancestral activity consisting of the seasonal movement of the livestock livestock over large long distances to reach warmer regions during the winter and together with a return to the mountains in summer 130 where pastures are greener and more suitable for grazing activities (Ruiz Perez and Valero Sáez, 1990; López Sáez et al., 2009). Particularly, the The livestock livesleaves the GIC3 region to go to southern Spanish regions, such as Extremadura, during the cold period. The enteric fermentation due to digestive processes in animals could thus be a significant CH4 source in this area. The Unión de Pequeños Agricultores (UPA, 2009) reports that between 2004 and 2009 an average of 800,000 transhumant 135 animals were hosted in Spain and 40,000 (5% of total) were counted in the province of Ávila (extension: 8,048 km2), where the GIC3 station is located. According to the available literature, in this area 85% of livestock still performs transhumance, with 500 stockbreeders moving every winter from the Gredos Natural Park (GNP) to warmer areas of Spain, such as Extremadura (Ruiz Perez and Valero Sáez, 1990; López Sáez et al., 2009; Libro Blanco, 2013). Generally, this mobility of the cattle and its associated  $CH_4$ 140 emissions (i.e. a major regional CH4 source) cannot easily be included in country-wide (annual) bottomup-inventories because it ishas not vet been properly quantified and reported by nations. The present study wantsaims to highlight the utility of 222Rn as a tracer to retrieve independent GHGs fluxes on a monthly basis using atmospheric 222Rn and CH4 concentrations-data. This work represents a first step towardtowards a better further characterization of "mobile"transient sources, such as transhumant

145 livestock for CH4, which could help to improve national emissions inventories. Finally, it offers new CH4 data for an under-sampled area which will help in the improvement of the regional and global methane budgets.

150

GIC3 is a new atmospheric station thusso its location, the surrounding region and the instrumentation used at this station have beenare described in the methodology section of this manuscript. In the first part of the results section both the daily and seasonal changes in  $CH_4$  concentrationsmixing ratios observed at the GIC3 station have been analysed in relation to 222Rn and PBLH variability. In the second part, the localnocturnal  $CH_4$  fluxes and their monthly variability have been estimated by the Radon Tracer Method

(RTM), following Vogel et al. (2012), and using an emission inventory for CH4 (EDGARv4.2). Both

- 155 sourceflux estimation methods have been applied taking into accountusing the same source region as modelled by the atmospheric transport model FLEXPARTv6FLEXPARTv9.0.2. The possible influence of biglarge cities surrounding GIC3 and of seasonally changing meteorological conditions on the retrieved CH4 fluxes has also been investigated. Finally, the difference in CH4 fluxes between the warmCattle season, defined by the presence of thewhen livestock is present in the GIC3 region, and the coldNo-Cattle
- 160 season, when the transhumant cattle migrateshave migrated to the south of Spain, calculated using the RTM, has been estimated.

**2 Methods**

**2.1 Study site: Gredos and Iruelas station (GIC3)**

The Gredos and Iruelas station (GIC3) is located in a rural region of the Spanish central plateau (40.35°N; 5.17°E; 1440 m above sea level (a.s.l.-,.\_)), as shown in Figure 1)-S1 of the supplement. GIC3 is set inlocated on the west side of the Gredos NaturalNational Park (GNP), which has a total extension of 86,397 ha. The mountains of the GNP form the highest mountain range in the E-W orientated central mountain system that divides the Iberian Peninsula in two parts. The GNP is located inhas a, predominantly, granitic basement; this type of and is thus covered by soil presentswith high activity levels of 228U (Nazaroff and Nero, 1988). The average 222Rn flux in this area is of about 70-100 Bq m-2 h-1 (e.g., López-Coto et al., 2013; -Karstens et al., -2015)), which is almost twice the average radon flux in central Europe (Szegvary et al., 2009, López-Coto et al., 2013; Grossi et al., 2016). The vegetation atin the GIC3 area is stratified according to the altitude and the main land use practice is a mixture of agro-forestry

175 exploitation (Figure 1). EEA, 2013)

CORINE land cover map 2006 for Spain with GIC3 (star label) and surrounding large cities Figure 1. Madrid, Salamanca, Valladolid and Avila).

180

Particularly, livestock farming is one of the main economic activities in the area around the GIC3 station (Ruiz Perez and Valero Sáez, 1990; López Saéz et al., 2009; MMA, 2016; Hernández, significant CH4 2016). The enteric fermentation due to digestive processes in animal can thus e in this area. The Unión de Pequeños Agricultores (UPA, 2009) reports that between 2004 and 2009 185 an average of 800,000 transhumant animals were hosted in Spain and 40,000 (5% of total) were counted province of Ávila, where GIC3 station is located. According to the available literature, in this area <del>of livestock still performs transhumance, with 500 stockbreeders moving every winter from the</del> GNPIn the GNP the seasonal migration of livestock starts between November and December-to-warmer areas of Spain, such as Extremadura (Ruiz Perez and Valero Sáez, 1990; López Sáez et al., 2009; Libro Blanco, 2013). In the GNP the seasonal migration of livestock starts in early November, when they travel 190 to the south of the Iberian Peninsula, and they do not return until late May-mid June (Ruiz Perez and Valero Sáez, 1990). In Figure S1S2 of the supplement, a map of the main Spanish transhumant paths is presented. The path used by the livestock present at GIC3 region is presented as a zoom-in subplot, indicating the entrance location (Puerto del Pico). Unfortunately, no specific reports with data about the 195 mobility rate of cattle or a local livestock count for individual months of the year mobility data are not so far available for the GIC3 area.

200

Besides livestock activities, there are three small-sized to medium-sized water reservoirs and four medium-sizesized to large cities in the wider area surrounding GIC3. Several The water reservoirs as well as several activities present in these the cities, e.g. landfills or waste water treatment plants, represent  $CH_4$ sources which could also influence methane concentrations observed at the GIC3 station under specific Formatted: Internet Link, Font: 12 pt

[revised manuscript text omitted]

| rediscende cheamations with as analysis and climate models and cheaved that these latter two modess                         | 1 |                                                                           |
|-----------------------------------------------------------------------------------------------------------------------------|---|---------------------------------------------------------------------------|
| deeper layers due to the difficulty in simulating stable conditions                                                         |   | Formatted: Font: Bold                                                     |
| -                                                                                                                           |   | Formatted. Font. Bold                                                     |
|                                                                                                                             |   |                                                                           |
|                                                                                                                             |   | Formatted: Font: 10 pt. Bold. English                                     |
| 2.4 CH 4 fluxes                                                                                                  | [ | (U.K.)                                                                    |
|                                                                                                                             |   | Formatted: Font: 10 pt, English (U.K.)                                    |
| 2.4.1 2.4.1 FLEXPART_RTM_CH₄ fluxes based on FLEXPART footprints and the Radon ←                                     |   | Formatted: Outline numbered +                                             |
| Tracer Method                                                                                                               |   | Level: 3 + Numbering Style: 1, 2, 3,
+ Start at: 1 + Alignment: Left + |
| · · · · · · · · · · · · · · · · · · ·                                                                                       |   | Aligned at: 0 cm + Tab after: 0 cm +                                      |
| The RTM is a well-known method (e.g. Hammer and Levin 2009) and it has been used in this study,                             |   | Formatted: Font: 10 pt, English (U.K.)                                    |
| following the implementation described in Vogel et al. (2012) in order to obtain observation-based                          |   |                                                                           |
| estimates of the nocturnal $CH_4$ fluxes at GIC3. The RTM uses atmospheric measurements of $^{222}$ Rn and                  |   |                                                                           |
| measured, or modelled, values of its-222Rn fluxes together with atmospheric concentrationsmixing ratios                     |   |                                                                           |
| of ana gas of interest-gas, i.e. CH 4 , in order to retrieve the net fluxes of this gas (e.g. Hammer and Levin   |   |                                                                           |
| 2009; Grossi et al., 2014).                                                                                                 |   |                                                                           |
| -                                                                                                                           |   |                                                                           |
| This method is based on the main assumption that the nocturnal lower atmospheric                                            |   | Formatted: English (U.S.)                                                 |
| boundary layer can be described as a well-mixed box of air (Schmidt et al. 1996; Levin                                      |   | Formatted: English (U.S.)                                                 |
| et al., 2011: Vogel et al., 2012). In this atmospheric box the variation of the concentration of any                        |   | Formatted: English (U.S.)                                                 |
| tracer with time $C(t)$ will be proportional to the flux of the tracer $F(t)$ and inversely proportional to the             |   |                                                                           |
| height of the boundary laver (h.(t)) (Eq.1: e.g. Griffiths et al., 2012; Grossi et al., 2014).                              |   | Formatted: English (U.S.)                                                 |
|                                                                                                                             | Γ |                                                                           |
|                                                                                                                             |   |                                                                           |
|                                                                                                                             |   |                                                                           |
| <del>(1)</del>                                                                                                              |   |                                                                           |
| The boundary layer is considered homogeneous within the box and with a time varying height. No                              | l |                                                                           |
| significant horizontal advection is considered due to stable atmospheric conditions (Griffiths et al., 2012).               |   | Formatted: Not Highlight                                                  |
| In this atmospheric volume the variation of the concentration of any tracer (shown with the subindex i)                     |   |                                                                           |
| with time $C_i(t)$ will be proportional to the flux of the tracer $F_i(t)$ and inversely proportional to the height         |   |                                                                           |
| of the boundary layer h(t) (Eq.1; e.g. Galmarini, 2006; Griffiths et al., 2012; Vogel et al., 2012; Grossi et               |   |                                                                           |
| al., 2014)                                                                                                           |   | Formatted: English (U.S.)                                                 |
|                                                                                                                             |   |                                                                           |
| $\frac{\mathrm{d}C_i(t)}{\mathrm{d}C_i(t)} \propto F_i(t) \cdot \frac{1}{\mathrm{d}C_i(t)} \tag{1}$                         |   |                                                                           |
| dt $h(t)$                                                                                                                   |   |                                                                           |
| Applying Eq. 1 for both 222 Rn and CH 4 , Eq. 2 is obtained, with a dimensionless conversion factor c |   |                                                                           |
| derived from the observed slope of the concurrent concentration increase of both gases:                                     |   |                                                                           |
|                                                                                                                             | I | Formatted: Footer Right                                                   |
| 9~                                                                                                                          |   | - Childreed. 1 ooter, Night                                               |
|                                                                                                                             | l |                                                                           |

$$\frac{\frac{dc_{CH_4}(t)}{dt}}{\frac{dc_{222}R_n(t)}{dt}} \cdot F_{222}R_n = c \cdot F_{222}R_n = FR_CH_4$$
(2).

Observing the concentration increase of two gases that fulfil the above assumptions, here CH4 and 222Rn7 and knowing. If the flux of 222Rnis known then the flux of CH4 can be calculated (Levin et al., 2011). A description of the specific criteria used to implement the RTM, which include selection criteria to reject situations with unstable atmospheric conditions, remote influences on the concentration and outliers detection, can be found in detail in Vogel et al. (2012). Grossi et al. (2014) previously applied the RTM for the first time at the GIC3 station using only a 3-monthsmonth dataset and with a constant (in time and space) 222Rn flux7 of 60 Bq m-2 h-1. Here, in order to apply the RTM to retrieve a time series of CH4 fluxes (FLEXPART\_RTMFR\_CH4) during 2013-2015 at the GIC3 station and to compare these results with the onesthose obtained using a bottom-up inventory for methane (FLEXPART\_EDGARFE\_CH4), we used the following extensive setupset-up;

330

340

 A nocturnal window between 20.00 UTC and 05.00 UTC was selected for theeach single night analysis in order to utilize only accumulation events when atmospheric concentrations of both CH4 and 222Rn had a positive concentration gradient due to positive net fluxes under stable boundary layer conditions;

335 2. A data selection criterion based on a threshold of  $R^2 \ge 0.8$  for the linear correlation between 222Rn and CH4 was used to reject events with low linear correlation between the atmospheric concentrations of both gases;

3. An *effective* local radon flux influencing the\_GIC3 station each night from 2013 to 2015 was calculated by\_coupling local radon flux data, obtained using the UHU modeloutput for the local pixel containing the GIC3 station of the model (developed by López-Coto et al., (2013), with the footprints calculated by ECMWF-FLEXPART model (version 69.02) (Stohl, 1998). RadonLocal radon flux data were calculated as explained in the following paragraph-and, while the footprints obtained are described in sectionSection 2.4.3,

The radon flux model (of Huelva University (from now on named the UHU model) employed in this work has been described in detail by López-Coto et al. (2013).-By using this model, a time-dependent inventory was calculated for the period 2011–2014 by employing several dynamic inputs, namely soil moisture, soil temperature-and snow cover thickness. These data were obtained directly from Weather Research and Forecasting (WRF) simulations (Skamarock et al., 2008). A domain of 97 x 97 grid cells centered incentred on Spain with a spatial resolution for grid of 0.2 degrees of 27 x 27 km2 and a temporal resolution of 1 hour-was defined (López Coto et al., 2013).- 2222Rn flux data calculated using this model were only available until November 2014 due to a lack of WRF simulations. In order to fulfil theobtain data for this period when modelled 222Rn flux data were not available, from December 2014 to December 2015, a seasonal and monthly climatology was calculated by using the UHU data set of UHU-model for the years 2011-2014. Karstens et al. (2015) compared the 222Rn flux values calculated over Europe by

| Formatted: Font: 10 pt, English (U.K.)                |
|-------------------------------------------------------|
| Formatted: Font: 10 pt, English (U.S.)                |
| Formatted: Font: 10 pt, English (U.K.)                |
| Formatted: Font color: Custom
Color(RGB(34,34,34)) |
|                                                       |

their model to UHU values and to long-term direct measurements of 222Rn exhalation rates in different areas of Europe. They found a generally 40% higher 222Rn exhalation rate on their map than estimated by the UHU map over Europe. This previous result has been taken into consideration within the present study to better interpret the obtained data.

360

**2.4.2 FLEXPART\_EDGAR\_CH4 fluxes**

**2.4.2 CH4 fluxes based on FLEXPART footprints and the EDGARv4.2 inventory grid map**

- 365 Bottom-up CH4 fluxes influencing the GIC3 station were estimated by using the footprints calculated by the ECMWF-FLEXPART model (obtained as described in sectionSection 2.4.3) and the Emissions Database for Global Atmospheric Research (EDGAR) version 4.2 (EDGAR, 2010). The EDGAR inventory, developed by the European Commission-Joint Research Centre- and the Netherlands Environmental Assessment Agency, includes global anthropogenic emissions of GHGs and air
- 370 pollutants by country and on a spatial grid. The EDGAR version used in the present study provides spatial (cells of 0.1 degree)global annual mean CH4 emissions globallyon a 0.1 degree (11 km) resolution for the year 2010. All major anthropogenic source sectors, e.g. waste treatment, industrial and agricultural sources (e.g. enteric fermentation) are included, whereas natural sources (e.g. wetlands or rivers) are not. The spatial allocation of emissions on 0.1 degree by 0.1 degree grid cells in EDGAR has been built up by
- 375 using spatial proxy datasets with the location of energy and manufacturing facilities, road networks, shipping routes, human and animal population density and agricultural land use. UNFCCC reported national sector totals are then removed with the given percentages of the spatial proxies over the country's area (EDGAR, 2010). Figure 1 shows the EDGAR inventory grid map extracted for Spain,
- 380 The influence of the emissions associated towith the cities surrounding the region of GIC3 was also modelled using this inventory to better understand their impact. In Table S1 of the supplement the coordinates of the upper and lower corners of the areas used to describe the location of the metropolitan areas over the EDGAR inventory are reported.

| Formatted: | Font: | 10 pt | , English | (U.K.) |
|------------|-------|-------|-----------|--------|
|            |       |       |           |        |
| Formatted: | Font: | Not E | Bold      |        |